# Toward Deeper Understanding of Neural Networks: The Power of Initialization and a Dual View on Expressivity

**Amit Daniely**
Google Brain

**Roy Frostig**[*]
Google Brain

**Yoram Singer**
Google Brain

## Abstract

We develop a general duality between neural networks and compositional kernel Hilbert spaces. We introduce the notion of a computation skeleton, an acyclic graph that succinctly describes both a family of neural networks and a kernel space. Random neural networks are generated from a skeleton through node replication followed by sampling from a normal distribution to assign weights. The kernel space consists of functions that arise by compositions, averaging, and non-linear transformations governed by the skeleton's graph topology and activation functions. We prove that random networks induce representations which approximate the kernel space. In particular, it follows that random weight initialization often yields a favorable starting point for optimization despite the worst-case intractability of training neural networks.

## 1   Introduction

Neural network (NN) learning has underpinned state of the art empirical results in numerous applied machine learning tasks, see for instance [25, 26]. Nonetheless, theoretical analyses of neural network learning are still lacking in several regards. Notably, it remains unclear why training algorithms find good weights and how learning is impacted by network architecture and its activation functions.

This work analyzes the representation power of neural networks within the vicinity of random initialization. We show that for regimes of practical interest, randomly initialized neural networks well-approximate a rich family of hypotheses. Thus, despite worst-case intractability of training neural networks, commonly used initialization procedures constitute a favorable starting point for training.

Concretely, we define a *computation skeleton* that is a succinct description of feed-forward networks. A skeleton induces a family of network architectures as well as an hypothesis class $\mathcal{H}$ of functions obtained by non-linear compositions mandated by the skeleton's structure. We then analyze the set of functions that can be expressed by varying the weights of the last layer, a simple region of the training domain over which the objective is convex. We show that with high probability over the choice of initial network weights, any function in $\mathcal{H}$ can be approximated by selecting the final layer's weights. Before delving into technical detail, we position our results in the context of previous research.

**Current theoretical understanding of NN learning.**   Standard results from complexity theory [22] imply that all efficiently computable functions can be expressed by a network of moderate size. Barron's theorem [7] states that even two-layer networks can express a very rich set of functions. The generalization ability of algorithms for training neural networks is also fairly well studied. Indeed, both classical [3, 9, 10] and more recent [18, 33] results from statistical learning theory show that, as the number of examples grows in comparison to the size of the network, the empirical risk approaches the population risk. In contrast, it remains puzzling why and when efficient algorithms, such as stochastic gradient methods, yield solutions that perform well. While learning algorithms succeed in

---

[*]Most of this work performed while the author was at Stanford University.

practice, theoretical analyses are overly pessimistic. For example, hardness results suggest that, in the worst case, even very simple 2-layer networks are intractable to learn. Concretely, it is hard to construct a hypothesis which predicts marginally better than random [15, 23, 24].

In the meantime, recent empirical successes of neural networks prompted a surge of theoretical results on NN learning. For instance, we refer the reader to [1, 4, 12, 14, 16, 28, 32, 38, 42] and the references therein.

**Compositional kernels and connections to networks.** The idea of composing kernels has repeatedly appeared in the machine learning literature. See for instance the early work by Grauman and Darrell [17], Schölkopf et al. [41]. Inspired by deep networks' success, researchers considered deep composition of kernels [11, 13, 29]. For fully connected two-layer networks, the correspondence between kernels and neural networks with random weights has been examined in [31, 36, 37, 45]. Notably, Rahimi and Recht [37] proved a formal connection, in a similar sense to ours, for the RBF kernel. Their work was extended to include polynomial kernels [21, 35] as well as other kernels [5, 6]. Several authors have further explored ways to extend this line of research to deeper, either fully-connected networks [13] or convolutional networks [2, 20, 29].

This work establishes a common foundation for the above research and expands the ideas therein. We extend the scope from fully-connected and convolutional networks to a broad family of architectures. In addition, we prove approximation guarantees between a network and its corresponding kernel in our general setting. We thus generalize previous analyses which are only applicable to fully connected two-layer networks.

## 2 Setting

**Notation.** We denote vectors by bold-face letters (e.g. $\mathbf{x}$), and matrices by upper case Greek letters (e.g. $\Sigma$). The 2-norm of $\mathbf{x} \in \mathbb{R}^d$ is denoted by $\|\mathbf{x}\|$. For functions $\sigma : \mathbb{R} \to \mathbb{R}$ we let

$$\|\sigma\| := \sqrt{\mathbb{E}_{X \sim \mathcal{N}(0,1)} \sigma^2(X)} = \sqrt{\frac{1}{\sqrt{2\pi}} \int_{-\infty}^{\infty} \sigma^2(x) e^{-\frac{x^2}{2}} dx} \, .$$

Let $G = (V, E)$ be a directed acyclic graph. The set of neighbors incoming to a vertex $v$ is denoted

$$\text{in}(v) := \{u \in V \mid uv \in E\} \, .$$

The $d-1$ dimensional sphere is denoted $\mathbb{S}^{d-1} = \{\mathbf{x} \in \mathbb{R}^d \mid \|\mathbf{x}\| = 1\}$. We provide a brief overview of reproducing kernel Hilbert spaces in the sequel and merely introduce notation here. In a Hilbert space $\mathcal{H}$, we use a slightly non-standard notation $\mathcal{H}^B$ for the ball of radius $B$, $\{\mathbf{x} \in \mathcal{H} \mid \|\mathbf{x}\|_{\mathcal{H}} \leq B\}$. We use $[x]_+$ to denote $\max(x, 0)$ and $\mathbf{1}[b]$ to denote the indicator function of a binary variable $b$.

**Input space.** Throughout the paper we assume that each example is a sequence of $n$ elements, each of which is represented as a unit vector. Namely, we fix $n$ and take the input space to be $\mathcal{X} = \mathcal{X}_{n,d} = \left(\mathbb{S}^{d-1}\right)^n$. Each input example is denoted,

$$\mathbf{x} = (\mathbf{x}^1, \ldots, \mathbf{x}^n), \text{ where } \mathbf{x}^i \in \mathbb{S}^{d-1} \, . \tag{1}$$

We refer to each vector $\mathbf{x}^i$ as the input's $i$th *coordinate*, and use $x_j^i$ to denote it $j$th scalar entry. Though this notation is slightly non-standard, it unifies input types seen in various domains. For example, binary features can be encoded by taking $d = 1$, in which case $\mathcal{X} = \{\pm 1\}^n$. Meanwhile, images and audio signals are often represented as bounded and continuous numerical values; we can assume in full generality that these values lie in $[-1, 1]$. To match the setup above, we embed $[-1, 1]$ into the circle $\mathbb{S}^1$, e.g. through the map

$$x \mapsto \left( \sin\left(\frac{\pi x}{2}\right), \cos\left(\frac{\pi x}{2}\right) \right) \, .$$

When each coordinate is categorical, taking one of $d$ values, one can represent the category $j \in [d]$ by the unit vector $\mathbf{e}_j \in \mathbb{S}^{d-1}$. When $d$ is very large or the basic units exhibit some structure—such as when the input is a sequence of words—a more concise encoding may be useful, e.g. using unit vectors in a low dimension space $\mathbb{S}^{d'}$ where $d' \ll d$ (see for instance Levy and Goldberg [27], Mikolov et al. [30]).

**Supervised learning.** The goal in supervised learning is to devise a mapping from the input space $\mathcal{X}$ to an output space $\mathcal{Y}$ based on a sample $S = \{(\mathbf{x}_1, y_1), \ldots, (\mathbf{x}_m, y_m)\}$, where $(\mathbf{x}_i, y_i) \in \mathcal{X} \times \mathcal{Y}$, drawn i.i.d. from a distribution $\mathcal{D}$ over $\mathcal{X} \times \mathcal{Y}$. A supervised learning problem is further specified by an output length $k$ and a loss function $\ell : \mathbb{R}^k \times \mathcal{Y} \to [0, \infty)$, and the goal is to find a predictor $h : \mathcal{X} \to \mathbb{R}^k$ whose loss,

$$\mathcal{L}_{\mathcal{D}}(h) := \mathop{\mathbb{E}}_{(\mathbf{x},y) \sim \mathcal{D}} \ell(h(\mathbf{x}), y)$$

is small. The *empirical* loss

$$\mathcal{L}_S(h) := \frac{1}{m} \sum_{i=1}^{m} \ell(h(\mathbf{x}_i), y_i)$$

is commonly used as a proxy for the loss $\mathcal{L}_{\mathcal{D}}$. Regression problems correspond to $\mathcal{Y} = \mathbb{R}$ and, for instance, the squared loss $\ell(\hat{y}, y) = (\hat{y} - y)^2$. Binary classification is captured by $\mathcal{Y} = \{\pm 1\}$ and, say, the zero-one loss $\ell(\hat{y}, y) = \mathbf{1}[\hat{y}y \leq 0]$ or the hinge loss $\ell(\hat{y}, y) = [1 - \hat{y}y]_+$, with standard extensions to the multiclass case. A loss $\ell$ is $L$-Lipschitz if $|\ell(y_1, y) - \ell(y_2, y)| \leq L|y_1 - y_2|$ for all $y_1, y_2 \in \mathbb{R}^k$, $y \in \mathcal{Y}$, and it is convex if $\ell(\cdot, y)$ is convex for every $y \in \mathcal{Y}$.

**Neural network learning.** We define a *neural network* $\mathcal{N}$ to be directed acyclic graph (DAG) whose nodes are denoted $V(\mathcal{N})$ and edges $E(\mathcal{N})$. Each of its internal units, i.e. nodes with both incoming and outgoing edges, is associated with an *activation* function $\sigma_v : \mathbb{R} \to \mathbb{R}$. In this paper's context, an activation can be any function that is square integrable with respect to the Gaussian measure on $\mathbb{R}$. We say that $\sigma$ is *normalized* if $\|\sigma\| = 1$. The set of nodes having only incoming edges are called the output nodes. To match the setup of a supervised learning problem, a network $\mathcal{N}$ has $nd$ input nodes and $k$ output nodes, denoted $o_1, \ldots, o_k$. A network $\mathcal{N}$ together with a weight vector $\mathbf{w} = \{w_{uv} \mid uv \in E\}$ defines a predictor $h_{\mathcal{N}, \mathbf{w}} : \mathcal{X} \to \mathbb{R}^k$ whose prediction is given by "propagating" $\mathbf{x}$ forward through the network. Formally, we define $h_{v, \mathbf{w}}(\cdot)$ to be the output of the subgraph of the node $v$ as follows: for an input node $v$, $h_{v, \mathbf{w}}$ is the identity function, and for all other nodes, we define $h_{v, \mathbf{w}}$ recursively as

$$h_{v, \mathbf{w}}(\mathbf{x}) = \sigma_v \left( \sum_{u \in \mathrm{in}(v)} w_{uv} \, h_{u, \mathbf{w}}(\mathbf{x}) \right) .$$

Finally, we let $h_{\mathcal{N}, \mathbf{w}}(\mathbf{x}) = (h_{o_1, \mathbf{w}}(\mathbf{x}), \ldots, h_{o_k, \mathbf{w}}(\mathbf{x}))$. We also refer to internal nodes as *hidden units*. The *output layer* of $\mathcal{N}$ is the sub-network consisting of all output neurons of $\mathcal{N}$ along with their incoming edges. The *representation* induced by a network $\mathcal{N}$ is the network $\mathrm{rep}(\mathcal{N})$ obtained from $\mathcal{N}$ by removing the output layer. The *representation function* induced by the weights $\mathbf{w}$ is $\mathcal{R}_{\mathcal{N}, \mathbf{w}} := h_{\mathrm{rep}(\mathcal{N}), \mathbf{w}}$. Given a sample $S$, a learning algorithm searches for weights $\mathbf{w}$ having small empirical loss $\mathcal{L}_S(\mathbf{w}) = \frac{1}{m} \sum_{i=1}^{m} \ell(h_{\mathcal{N}, \mathbf{w}}(\mathbf{x}_i), y_i)$. A popular approach is to randomly initialize the weights and then use a variant of the stochastic gradient method to improve these weights in the direction of lower empirical loss.

**Kernel learning.** A function $\kappa : \mathcal{X} \times \mathcal{X} \to \mathbb{R}$ is a *reproducing kernel*, or simply a kernel, if for every $\mathbf{x}_1, \ldots, \mathbf{x}_r \in \mathcal{X}$, the $r \times r$ matrix $\Gamma_{i,j} = \{\kappa(\mathbf{x}_i, \mathbf{x}_j)\}$ is positive semi-definite. Each kernel induces a Hilbert space $\mathcal{H}_\kappa$ of functions from $\mathcal{X}$ to $\mathbb{R}$ with a corresponding norm $\|\cdot\|_{\mathcal{H}_\kappa}$. A kernel and its corresponding space are *normalized* if $\forall \mathbf{x} \in \mathcal{X}$, $\kappa(\mathbf{x}, \mathbf{x}) = 1$. Given a convex loss function $\ell$, a sample $S$, and a kernel $\kappa$, a kernel learning algorithm finds a function $f = (f_1, \ldots, f_k) \in \mathcal{H}_\kappa^k$ whose empirical loss, $\mathcal{L}_S(f) = \frac{1}{m} \sum_i \ell(f(\mathbf{x}_i), y_i)$, is minimal among all functions with $\sum_i \|f_i\|_\kappa^2 \leq R^2$ for some $R > 0$. Alternatively, kernel algorithms minimize the *regularized loss*,

$$\mathcal{L}_S^R(f) = \frac{1}{m} \sum_{i=1}^{m} \ell(f(\mathbf{x}_i), y_i) + \frac{1}{R^2} \sum_{i=1}^{k} \|f_i\|_\kappa^2 ,$$

a convex objective that often can be efficiently minimized.

## 3 Computation skeletons

In this section we define a simple structure that we term a computation skeleton. The purpose of a computational skeleton is to compactly describe feed-forward computation from an input to an output. A single skeleton encompasses a family of neural networks that share the same skeletal structure. Likewise, it defines a corresponding kernel space.

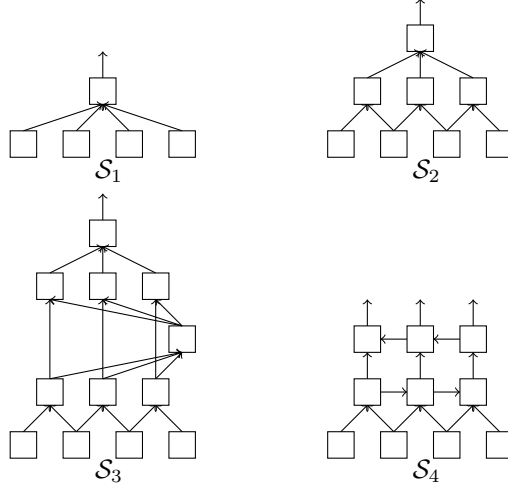

Figure 1: Examples of computation skeletons.

**Definition.** *A computation skeleton $\mathcal{S}$ is a DAG whose non-input nodes are labeled by activations.*

Though the formal definition of neural networks and skeletons appear identical, we make a conceptual distinction between them as their role in our analysis is rather different. Accompanied by a set of weights, a neural network describes a concrete function, whereas the skeleton stands for a topology common to several networks as well as for a kernel. To further underscore the differences we note that skeletons are naturally more compact than networks. In particular, all examples of skeletons in this paper are *irreducible*, meaning that for each two nodes $v, u \in V(\mathcal{S})$, $\text{in}(v) \neq \text{in}(u)$. We further restrict our attention to skeletons with a single output node, showing later that single-output skeletons can capture supervised problems with outputs in $\mathbb{R}^k$. We denote by $|\mathcal{S}|$ the number of non-input nodes of $\mathcal{S}$.

Figure 1 shows four example skeletons, omitting the designation of the activation functions. The skeleton $\mathcal{S}_1$ is rather basic as it aggregates all the inputs in a single step. Such topology can be useful in the absence of any prior knowledge of how the output label may be computed from an input example, and it is commonly used in natural language processing where the input is represented as a bag-of-words [19]. The only structure in $\mathcal{S}_1$ is a single *fully connected* layer:

**Terminology** (Fully connected layer of a skeleton). *An induced subgraph of a skeleton with $r + 1$ nodes, $u_1, \ldots, u_r, v$, is called a* fully connected *layer if its edges are $u_1 v, \ldots, u_r v$.*

The skeleton $\mathcal{S}_2$ is slightly more involved: it first processes consecutive (overlapping) parts of the input, and the next layer aggregates the partial results. Altogether, it corresponds to networks with a single one-dimensional convolutional layer, followed by a fully connected layer. The two-dimensional (and deeper) counterparts of such skeletons correspond to networks that are common in visual object recognition.

**Terminology** (Convolution layer of a skeleton). *Let $s, w, q$ be positive integers and denote $n = s(q-1) + w$. A subgraph of a skeleton is a one dimensional* convolution layer *of width $w$ and stride $s$ if it has $n + q$ nodes, $u_1, \ldots, u_n, v_1, \ldots, v_q$, and $qw$ edges, $u_{s(i-1)+j} v_i$, for $1 \leq i \leq q, 1 \leq j \leq w$.*

The skeleton $\mathcal{S}_3$ is a somewhat more sophisticated version of $\mathcal{S}_2$: the local computations are first aggregated, then reconsidered with the aggregate, and finally aggregated again. The last skeleton, $\mathcal{S}_4$, corresponds to the networks that arise in learning sequence-to-sequence mappings as used in translation, speech recognition, and OCR tasks (see for example Sutskever et al. [44]).

## 3.1 From computation skeletons to neural networks

The following definition shows how a skeleton, accompanied with a replication parameter $r \geq 1$ and a number of output nodes $k$, induces a neural network architecture. Recall that inputs are ordered sets of vectors in $\mathbb{S}^{d-1}$.

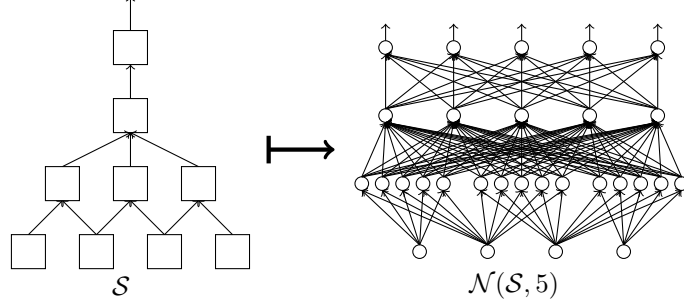

Figure 2: A 5-fold realizations of the computation skeleton $\mathcal{S}$ with $d = 1$.

**Definition** (Realization of a skeleton). *Let $\mathcal{S}$ be a computation skeleton and consider input coordinates in $\mathbb{S}^{d-1}$ as in (1). For $r, k \geq 1$ we define the following neural network $\mathcal{N} = \mathcal{N}(\mathcal{S}, r, k)$. For each input node in $\mathcal{S}$, $\mathcal{N}$ has $d$ corresponding input neurons. For each internal node $v \in \mathcal{S}$ labeled by an activation $\sigma$, $\mathcal{N}$ has $r$ neurons $v^1, \ldots, v^r$, each with an activation $\sigma$. In addition, $\mathcal{N}$ has $k$ output neurons $o_1, \ldots, o_k$ with the identity activation $\sigma(x) = x$. There is an edge $v^i u^j \in E(\mathcal{N})$ whenever $uv \in E(\mathcal{S})$. For every output node $v$ in $\mathcal{S}$, each neuron $v^j$ is connected to all output neurons $o_1, \ldots, o_k$. We term $\mathcal{N}$ the $(r, k)$-fold realization of $\mathcal{S}$. We also define the $r$-fold realization of $\mathcal{S}$ as[2] $\mathcal{N}(\mathcal{S}, r) = \mathrm{rep}\,(\mathcal{N}(\mathcal{S}, r, 1))$.*

Note that the notion of the replication parameter $r$ corresponds, in the terminology of convolutional networks, to the number of channels taken in a convolutional layer and to the number of hidden units taken in a fully-connected layer.

Figure 2 illustrates a 5-realization of a skeleton with coordinate dimension $d = 1$. The realization is a network with a single (one dimensional) convolutional layer having 5 channels, stride of 1, and width of 2, followed by two fully-connected layers. The global replication parameter $r$ in a realization is used for brevity; it is straightforward to extend results when the different nodes in $\mathcal{S}$ are each replicated to a different extent.

We next define a scheme for random initialization of the weights of a neural network, that is similar to what is often done in practice. We employ the definition throughout the paper whenever we refer to random weights.

**Definition** (Random weights). *A* random initialization *of a neural network $\mathcal{N}$ is a multivariate Gaussian $\mathbf{w} = (w_{uv})_{uv \in E(\mathcal{N})}$ such that each weight $w_{uv}$ is sampled independently from a normal distribution with mean 0 and variance $1/\left(\|\sigma_u\|^2 |\mathrm{in}(v)|\right)$.*

Architectures such as convolutional nets have weights that are shared across different edges. Again, it is straightforward to extend our results to these cases and for simplicity we assume no weight sharing.

### 3.2   From computation skeletons to reproducing kernels

In addition to networks' architectures, a computation skeleton $\mathcal{S}$ also defines a normalized kernel $\kappa_{\mathcal{S}} : \mathcal{X} \times \mathcal{X} \to [-1, 1]$ and a corresponding norm $\|\cdot\|_{\mathcal{S}}$ on functions $f : \mathcal{X} \to \mathbb{R}$. This norm has the property that $\|f\|_{\mathcal{S}}$ is small if and only if $f$ can be obtained by certain simple compositions of functions according to the structure of $\mathcal{S}$. To define the kernel, we introduce a *dual activation* and *dual kernel*. For $\rho \in [-1, 1]$, we denote by $N_\rho$ the multivariate Gaussian distribution on $\mathbb{R}^2$ with mean 0 and covariance matrix $\left(\begin{smallmatrix} 1 & \rho \\ \rho & 1 \end{smallmatrix}\right)$.

**Definition** (Dual activation and kernel). *The* dual activation *of an activation $\sigma$ is the function $\hat{\sigma} : [-1, 1] \to \mathbb{R}$ defined as*

$$\hat{\sigma}(\rho) = \underset{(X,Y) \sim N_\rho}{\mathbb{E}} \sigma(X)\sigma(Y).$$

*The* dual kernel *w.r.t. to a Hilbert space $\mathcal{H}$ is the kernel $\kappa_\sigma : \mathcal{H}^1 \times \mathcal{H}^1 \to \mathbb{R}$ defined as*

$$\kappa_\sigma(\mathbf{x}, \mathbf{y}) = \hat{\sigma}(\langle \mathbf{x}, \mathbf{y} \rangle_{\mathcal{H}}).$$

| Activation | | Dual Activation | Kernel | Ref |
|---|---|---|---|---|
| Identity | $x$ | $\rho$ | linear | |
| 2nd Hermite | $\frac{x^2-1}{\sqrt{2}}$ | $\rho^2$ | poly | |
| ReLU | $\sqrt{2}\,[x]_+$ | $\frac{1}{\pi} + \frac{\rho}{2} + \frac{\rho^2}{2\pi} + \frac{\rho^4}{24\pi} + \ldots = \frac{\sqrt{1-\rho^2}+(\pi-\cos^{-1}(\rho))\rho}{\pi}$ | $\arccos_1$ | [13] |
| Step | $\sqrt{2}\,\mathbf{1}[x \geq 0]$ | $\frac{1}{2} + \frac{\rho}{\pi} + \frac{\rho^3}{6\pi} + \frac{3\rho^5}{40\pi} + \ldots = \frac{\pi-\cos^{-1}(\rho)}{\pi}$ | $\arccos_0$ | [13] |
| Exponential | $e^{x-2}$ | $\frac{1}{e} + \frac{\rho}{e} + \frac{\rho^2}{2e} + \frac{\rho^3}{6e} + \ldots = e^{\rho-1}$ | RBF | [29] |

Table 1: Activation functions and their duals.

We show in the supplementary material that $\kappa_\sigma$ is indeed a kernel for every activation $\sigma$ that adheres with the square-integrability requirement. In fact, any continuous $\mu : [-1,1] \to \mathbb{R}$, such that $(\mathbf{x}, \mathbf{y}) \mapsto \mu(\langle \mathbf{x}, \mathbf{y} \rangle_{\mathcal{H}})$ is a kernel for all $\mathcal{H}$, is the dual of some activation. Note that $\kappa_\sigma$ is normalized iff $\sigma$ is normalized. We show in the supplementary material that dual activations are closely related to Hermite polynomial expansions, and that these can be used to calculate the duals of activation functions analytically. Table 1 lists a few examples of normalized activations and their corresponding dual (corresponding derivations are in supplementary material). The following definition gives the kernel corresponding to a skeleton having normalized activations.[3]

**Definition** (Compositional kernels). *Let $\mathcal{S}$ be a computation skeleton with normalized activations and (single) output node o. For every node $v$, inductively define a kernel $\kappa_v : \mathcal{X} \times \mathcal{X} \to \mathbb{R}$ as follows. For an input node $v$ corresponding to the ith coordinate, define $\kappa_v(\mathbf{x}, \mathbf{y}) = \langle \mathbf{x}^i, \mathbf{y}^i \rangle$. For a non-input node $v$, define*

$$\kappa_v(\mathbf{x}, \mathbf{y}) = \hat{\sigma}_v \left( \frac{\sum_{u \in \text{in}(v)} \kappa_u(\mathbf{x}, \mathbf{y})}{|\text{in}(v)|} \right) .$$

*The final kernel $\kappa_{\mathcal{S}}$ is $\kappa_o$, the kernel associated with the output node o. The resulting Hilbert space and norm are denoted $\mathcal{H}_{\mathcal{S}}$ and $\| \cdot \|_{\mathcal{S}}$ respectively, and $\mathcal{H}_v$ and $\| \cdot \|_v$ denote the space and norm when formed at node $v$.*

As we show later, $\kappa_{\mathcal{S}}$ is indeed a (normalized) kernel for every skeleton $\mathcal{S}$. To understand the kernel in the context of learning, we need to examine which functions can be expressed as moderate norm functions in $\mathcal{H}_{\mathcal{S}}$. As we show in the supplementary material, these are the functions obtained by certain simple compositions according to the feed-forward structure of $\mathcal{S}$. For intuition, we contrast two examples of two commonly used skeletons. For simplicity, we restrict to the case $\mathcal{X} = \mathcal{X}_{n,1} = \{\pm 1\}^n$, and omit the details of derivations.

**Example 1** (Fully connected skeletons). Let $\mathcal{S}$ be a skeleton consisting of $l$ fully connected layers, where the $i$'th layer is associated with the activation $\sigma_i$. We have $\kappa_{\mathcal{S}}(\mathbf{x}, \mathbf{x}') = \hat{\sigma}_l \circ \ldots \circ \hat{\sigma}_1 \left( \frac{\langle \mathbf{x}, \mathbf{y} \rangle}{n} \right)$. For such kernels, any moderate norm function in $\mathcal{H}$ is well approximated by a low degree polynomial. For example, if $\|f\|_{\mathcal{S}} \leq n$, then there is a second degree polynomial $p$ such that $\|f - p\|_2 \leq O\left( \frac{1}{\sqrt{n}} \right)$.

We next argue that convolutional skeletons define kernel spaces that are quite different from kernels spaces defined by fully connected skeletons. Concretely, suppose $f : \mathcal{X} \to \mathbb{R}$ is of the form $f = \sum_{i=1}^m f_i$ where each $f_i$ depends only on $q$ adjacent coordinates. We call $f$ a $q$-local function. In Example 1 we stated that for fully-connected skeletons, any function of in the induced space of norm less then $n$ is well approximated by second degree polynomials. In contrast, the following example underscores that for convolutional skeletons, we can find functions that are more complex, provided that they are local.

**Example 2** (Convolutional skeletons). Let $\mathcal{S}$ be a skeleton consisting of a convolutional layer of stride 1 and width $q$, followed by a single fully connected layer. (The skeleton $\mathcal{S}_2$ from Figure 1 is a concrete example with $q = 2$ and $n = 4$.) To simplify the argument, we assume that all activations are $\sigma(x) = e^x$ and $q$ is a constant. For any $q$-local function $f : \mathcal{X} \to \mathbb{R}$ we have

$$\|f\|_{\mathcal{S}} \leq C \cdot \sqrt{n} \cdot \|f\|_2 .$$

Here, $C > 0$ is a constant depending only on $q$. Hence, for example, any average of functions from $\mathcal{X}$ to $[-1, 1]$, each of which depends on $q$ adjacent coordinates, is in $\mathcal{H}_{\mathcal{S}}$ and has norm of merely $O(\sqrt{n})$.

## 4 Main results

We review our main results. Proofs can be found in the supplementary material. Let us fix a compositional kernel $\mathcal{S}$. There are a few upshots to underscore upfront. First, our analysis implies that a representation generated by a random initialization of $\mathcal{N} = \mathcal{N}(\mathcal{S}, r, k)$ approximates the kernel $\kappa_{\mathcal{S}}$. The sense in which the result holds is twofold. First, with the proper rescaling we show that $\langle \mathcal{R}_{\mathcal{N},\mathbf{w}}(\mathbf{x}), \mathcal{R}_{\mathcal{N},\mathbf{w}}(\mathbf{x}') \rangle \approx \kappa_{\mathcal{S}}(\mathbf{x}, \mathbf{x}')$. Then, we also show that the functions obtained by composing bounded linear functions with $\mathcal{R}_{\mathcal{N},\mathbf{w}}$ are approximately the bounded-norm functions in $\mathcal{H}_{\mathcal{S}}$. In other words, the functions expressed by $\mathcal{N}$ under varying the weights of the final layer are approximately bounded-norm functions in $\mathcal{H}_{\mathcal{S}}$. For simplicity, we restrict the analysis to the case $k = 1$. We also confine the analysis to either bounded activations, with bounded first and second derivatives, or the ReLU activation. Extending the results to a broader family of activations is left for future work. Through this and remaining sections we use $\gtrsim$ to hide universal constants.

**Definition.** *An activation* $\sigma : \mathbb{R} \to \mathbb{R}$ *is* $C$-bounded *if it is twice continuously differentiable and* $\|\sigma\|_\infty, \|\sigma'\|_\infty, \|\sigma''\|_\infty \leq \|\sigma\| C$.

Note that many activations are $C$-bounded for some constant $C > 0$. In particular, most of the popular sigmoid-like functions such as $1/(1 + e^{-x})$, $\mathrm{erf}(x)$, $x/\sqrt{1 + x^2}$, $\tanh(x)$, and $\tan^{-1}(x)$ satisfy the boundedness requirements. We next introduce terminology that parallels the representation layer of $\mathcal{N}$ with a kernel space. Concretely, let $\mathcal{N}$ be a network whose representation part has $q$ output neurons. Given weights $\mathbf{w}$, the *normalized representation* $\Psi_{\mathbf{w}}$ is obtained from the representation $R_{\mathcal{N},\mathbf{w}}$ by dividing each output neuron $v$ by $\|\sigma_v\|\sqrt{q}$. The *empirical kernel* corresponding to $\mathbf{w}$ is defined as $\kappa_{\mathbf{w}}(\mathbf{x}, \mathbf{x}') = \langle \Psi_{\mathbf{w}}(\mathbf{x}), \Psi_{\mathbf{w}}(\mathbf{x}') \rangle$. We also define the *empirical kernel space* corresponding to $\mathbf{w}$ as $\mathcal{H}_{\mathbf{w}} = \mathcal{H}_{\kappa_{\mathbf{w}}}$. Concretely,

$$\mathcal{H}_{\mathbf{w}} = \{h_{\mathbf{v}}(\mathbf{x}) = \langle \mathbf{v}, \Psi_{\mathbf{w}}(\mathbf{x}) \rangle \mid \mathbf{v} \in \mathbb{R}^q \} \ ,$$

and the norm of $\mathcal{H}_{\mathbf{w}}$ is defined as $\|h\|_{\mathbf{w}} = \inf\{\|\mathbf{v}\| \mid h = h_{\mathbf{v}}\}$. Our first result shows that the empirical kernel approximates the kernel $k_{\mathcal{S}}$.

**Theorem 3.** *Let* $\mathcal{S}$ *be a skeleton with* $C$-bounded *activations. Let* $\mathbf{w}$ *be a random initialization of* $\mathcal{N} = \mathcal{N}(\mathcal{S}, r)$ *with*

$$r \geq \frac{(4C^4)^{\mathrm{depth}(\mathcal{S})+1} \log\left(8|\mathcal{S}|/\delta\right)}{\epsilon^2} \ .$$

*Then, for all* $\mathbf{x}, \mathbf{x}' \in \mathcal{X}$*, with probability of at least* $1 - \delta$*,*

$$|k_{\mathbf{w}}(\mathbf{x}, \mathbf{x}') - k_{\mathcal{S}}(\mathbf{x}, \mathbf{x}')| \leq \epsilon \ .$$

We note that if we fix the activation and assume that the depth of $\mathcal{S}$ is logarithmic, then the required bound on $r$ is polynomial. For the ReLU activation we get a stronger bound with only quadratic dependence on the depth. However, it requires that $\epsilon \leq 1/\mathrm{depth}(\mathcal{S})$.

**Theorem 4.** *Let* $\mathcal{S}$ *be a skeleton with ReLU activations. Let* $\mathbf{w}$ *be a random initialization of* $\mathcal{N}(\mathcal{S}, r)$ *with*

$$r \gtrsim \frac{\mathrm{depth}^2(\mathcal{S}) \log\left(|\mathcal{S}|/\delta\right)}{\epsilon^2} \ .$$

*Then, for all* $\mathbf{x}, \mathbf{x}' \in \mathcal{X}$ *and* $\epsilon \lesssim 1/\mathrm{depth}(\mathcal{S})$*, with probability of at least* $1 - \delta$*,*

$$|\kappa_{\mathbf{w}}(\mathbf{x}, \mathbf{x}') - \kappa_{\mathcal{S}}(\mathbf{x}, \mathbf{x}')| \leq \epsilon \ .$$

For the remaining theorems, we fix a $L$-Lipschitz loss $\ell : \mathbb{R} \times \mathcal{Y} \to [0, \infty)$. For a distribution $\mathcal{D}$ on $\mathcal{X} \times \mathcal{Y}$ we denote by $\|\mathcal{D}\|_0$ the cardinality of the support of the distribution. We note that $\log\left(\|\mathcal{D}\|_0\right)$ is bounded by, for instance, the number of bits used to represent an element in $\mathcal{X} \times \mathcal{Y}$. We use the following notion of approximation.

**Definition.** *Let* $\mathcal{D}$ *be a distribution on* $\mathcal{X} \times \mathcal{Y}$*. A space* $\mathcal{H}_1 \subset \mathbb{R}^{\mathcal{X}}$ $\epsilon$-approximates *the space* $\mathcal{H}_2 \subset \mathbb{R}^{\mathcal{X}}$ *w.r.t.* $\mathcal{D}$ *if for every* $h_2 \in \mathcal{H}_2$ *there is* $h_1 \in \mathcal{H}_1$ *such that* $\mathcal{L}_{\mathcal{D}}(h_1) \leq \mathcal{L}_{\mathcal{D}}(h_2) + \epsilon$*.*

**Theorem 5.** *Let $\mathcal{S}$ be a skeleton with $C$-bounded activations and let $R > 0$. Let $\mathbf{w}$ be a random initialization of $\mathcal{N}(\mathcal{S}, r)$ with*

$$r \gtrsim \frac{L^4 \, R^4 \, (4C^4)^{\mathrm{depth}(\mathcal{S})+1} \log\left(\frac{LRC|\mathcal{S}|}{\epsilon\delta}\right)}{\epsilon^4} .$$

*Then, with probability of at least $1 - \delta$ over the choices of $\mathbf{w}$ we have that, for any data distribution, $\mathcal{H}_{\mathbf{w}}^{\sqrt{2}R}$ $\epsilon$-approximates $\mathcal{H}_{\mathcal{S}}^R$ and $\mathcal{H}_{\mathcal{S}}^{\sqrt{2}R}$ $\epsilon$-approximates $\mathcal{H}_{\mathbf{w}}^R$.*

**Theorem 6.** *Let $\mathcal{S}$ be a skeleton with ReLU activations, $\epsilon \lesssim 1/\mathrm{depth}(\mathcal{C})$, and $R > 0$. Let $\mathbf{w}$ be a random initialization of $\mathcal{N}(\mathcal{S}, r)$ with*

$$r \gtrsim \frac{L^4 \, R^4 \, \mathrm{depth}^2(\mathcal{S}) \, \log\left(\frac{\|\mathcal{D}\|_0|\mathcal{S}|}{\delta}\right)}{\epsilon^4} .$$

*Then, with probability of at least $1 - \delta$ over the choices of $\mathbf{w}$ we have that, for any data distribution, $\mathcal{H}_{\mathbf{w}}^{\sqrt{2}R}$ $\epsilon$-approximates $\mathcal{H}_{\mathcal{S}}^R$ and $\mathcal{H}_{\mathcal{S}}^{\sqrt{2}R}$ $\epsilon$-approximates $\mathcal{H}_{\mathbf{w}}^R$.*

As in Theorems 3 and 4, for a fixed $C$-bounded activation and logarithmically deep $\mathcal{S}$, the required bounds on $r$ are polynomial. Analogously, for the ReLU activation the bound is polynomial even without restricting the depth. However, the polynomial growth in Theorems 5 and 6 is rather large. Improving the bounds, or proving their optimality, is left to future work.

### Acknowledgments

We would like to thank Percy Liang and Ben Recht for fruitful discussions, comments, and suggestions.

## Footnotes

[2]Note that for every $k$, $\mathrm{rep}\,(\mathcal{N}(\mathcal{S}, r, 1)) = \mathrm{rep}\,(\mathcal{N}(\mathcal{S}, r, k))$.

[3] For a skeleton $\mathcal{S}$ with unnormalized activations, the corresponding kernel is the kernel of the skeleton $\mathcal{S}'$ obtained by normalizing the activations of $\mathcal{S}$.

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
