[Supplementary Material]

# A   Mathematical background

**Reproducing kernel Hilbert spaces (RKHS).**   The proofs of all the theorems we quote here are well-known and can be found in Chapter 2 of [39] and similar textbooks. Let $\mathcal{H}$ be a Hilbert space of functions from $\mathcal{X}$ to $\mathbb{R}$. We say that $\mathcal{H}$ is a *reproducing kernel Hilbert space*, abbreviated RKHS or kernel space, if for every $\mathbf{x} \in \mathcal{X}$ the linear functional $f \mapsto f(\mathbf{x})$ is bounded. The following theorem provides a one-to-one correspondence between kernels and kernel spaces.

**Theorem 7.** *(i) For every kernel $\kappa$ there exists a unique kernel space $\mathcal{H}_\kappa$ such that for every $\mathbf{x} \in \mathcal{X}$, $\kappa(\cdot, \mathbf{x}) \in \mathcal{H}_\kappa$ and for all $f \in \mathcal{H}_\kappa$, $f(\mathbf{x}) = \langle f(\cdot), \kappa(\cdot, \mathbf{x}) \rangle_{\mathcal{H}_\kappa}$.   (ii) A Hilbert space $\mathcal{H} \subseteq \mathbb{R}^{\mathcal{X}}$ is a kernel space if and only if there exists a kernel $\kappa : \mathcal{X} \times \mathcal{X} \to \mathbb{R}$ such that $\mathcal{H} = \mathcal{H}_\kappa$.*

The following theorem describes a tight connection between embeddings of $\mathcal{X}$ into a Hilbert space and kernel spaces.

**Theorem 8.** *A function $\kappa : \mathcal{X} \times \mathcal{X} \to \mathbb{R}$ is a kernel if and only if there exists a mapping $\Phi : \mathcal{X} \to \mathcal{H}$ to some Hilbert space for which $\kappa(\mathbf{x}, \mathbf{x}') = \langle \Phi(\mathbf{x}), \Phi(\mathbf{x}') \rangle_{\mathcal{H}}$. In addition, the following two properties hold,*

- $\mathcal{H}_\kappa = \{f_{\mathbf{v}} : \mathbf{v} \in \mathcal{H}\}$, *where* $f_{\mathbf{v}}(\mathbf{x}) = \langle \mathbf{v}, \Phi(\mathbf{x}) \rangle_{\mathcal{H}}$.

- *For every* $f \in \mathcal{H}_\kappa$, $\|f\|_{\mathcal{H}_\kappa} = \inf\{\|\mathbf{v}\|_{\mathcal{H}} \mid f = f_{\mathbf{v}}\}$.

**Positive definite functions.**   A function $\mu : [-1, 1] \to \mathbb{R}$ is *positive definite* (PSD) if there are non-negative numbers $b_0, b_1, \ldots$ such that

$$\sum_{i=0}^{\infty} b_i < \infty \;\text{ and }\; \forall x \in [-1, 1], \; \mu(x) = \sum_{i=0}^{\infty} b_i x^i \,.$$

The *norm* of $\mu$ is defined as $\|\mu\| := \sqrt{\sum_i b_i} = \sqrt{\mu(1)}$. We say that $\mu$ is *normalized* if $\|\mu\| = 1$

**Theorem 9** (Schoenberg, [40])**.**  *A continuous function $\mu : [-1, 1] \to \mathbb{R}$ is PSD if and only if for all $d = 1, 2, \ldots, \infty$, the function $\kappa : \mathbb{S}^{d-1} \times \mathbb{S}^{d-1} \to \mathbb{R}$ defined by $\kappa(\mathbf{x}, \mathbf{x}') = \mu(\langle \mathbf{x}, \mathbf{x}' \rangle)$ is a kernel.*

The restriction to the unit sphere of many of the kernels used in machine learning applications corresponds to positive definite functions. An example is the Gaussian kernel,

$$\kappa(\mathbf{x}, \mathbf{x}') = \exp\left(-\frac{\|\mathbf{x} - \mathbf{x}'\|^2}{2\sigma^2}\right) \,.$$

Indeed, note that for unit vectors $\mathbf{x}, \mathbf{x}'$ we have

$$\kappa(\mathbf{x}, \mathbf{x}') = \exp\left(-\frac{\|\mathbf{x}\|^2 + \|\mathbf{x}'\|^2 - 2\langle \mathbf{x}, \mathbf{x}' \rangle}{2\sigma^2}\right) = \exp\left(-\frac{1 - \langle \mathbf{x}, \mathbf{x}' \rangle}{\sigma^2}\right) \,.$$

Another example is the Polynomial kernel $\kappa(\mathbf{x}, \mathbf{x}') = \langle \mathbf{x}, \mathbf{x}' \rangle^d$.

**Hermite polynomials.**   The normalized *Hermite polynomials* is the sequence $h_0, h_1, \ldots$ of orthonormal polynomials obtained by applying the Gram-Schmidt process to the sequence $1, x, x^2, \ldots$ w.r.t. the inner-product $\langle f, g \rangle = \frac{1}{\sqrt{2\pi}} \int_{-\infty}^{\infty} f(x) g(x) e^{-\frac{x^2}{2}} dx$. Recall that we define activations as square integrable functions w.r.t. the Gaussian measure. Thus, Hermite polynomials form an orthonormal basis to the space of activations. In particular, each activation $\sigma$ can be uniquely described in the basis of Hermite polynomials,

$$\sigma(x) = a_0 h_0(x) + a_1 h_1(x) + a_2 h_2(x) + \ldots , \tag{2}$$

where the convergence holds in $\ell^2$ w.r.t. the Gaussian measure. This decomposition is called the Hermite *expansion*. Finally, we use the following facts (see Chapter 11 in [34] and the relevant entry

in Wikipedia):

$$\forall n \geq 1, \; h_{n+1}(x) \;=\; \frac{x}{\sqrt{n+1}} h_n(x) - \sqrt{\frac{n}{n+1}} h_{n-1}(x) \,, \tag{3}$$

$$\forall n \geq 1, \; h'_n(x) \;=\; \sqrt{n}\, h_{n-1}(x) \tag{4}$$

$$\mathop{\mathbb{E}}_{(X,Y)\sim N_\rho} h_m(X) h_n(Y) \;=\; \begin{cases} \rho^n & n = m \\ 0 & n \neq m \end{cases} \; \text{where} \; n, m \geq 0, \; \rho \in [-1, 1] \,, \tag{5}$$

$$h_n(0) \;=\; \begin{cases} 0, & \text{if } n \text{ is odd} \\ \frac{1}{\sqrt{n!}}(-1)^{\frac{n}{2}}(n-1)!! & \text{if } n \text{ is even} \end{cases} \,, \tag{6}$$

where

$$n!! = \begin{cases} 1 & n \leq 0 \\ n \cdot (n-2) \cdots 5 \cdot 3 \cdot 1 & n > 0 \text{ odd} \\ n \cdot (n-2) \cdots 6 \cdot 4 \cdot 2 & n > 0 \text{ even} \end{cases} .$$

## B  Compositional kernel spaces

We now describe the details of compositional kernel spaces. Let $\mathcal{S}$ be a skeleton with normalized activations and $n$ input nodes associated with the input's coordinates. Throughout the rest of the section we study the functions in $\mathcal{H}_\mathcal{S}$ and their norm. In particular, we show that $\kappa_\mathcal{S}$ is indeed a normalized kernel. Recall that $\kappa_\mathcal{S}$ is defined inductively by the equation,

$$\kappa_v(\mathbf{x}, \mathbf{x}') = \hat{\sigma}_v \left( \frac{\sum_{u \in \text{in}(v)} \kappa_u(\mathbf{x}, \mathbf{x}')}{|\text{in}(v)|} \right) . \tag{7}$$

The recursion (7) describes a means for generating a kernel form another kernel. Since kernels correspond to kernel spaces, it also prescribes an operator that produces a kernel space from other kernel spaces. If $\mathcal{H}_v$ is the space corresponding to $v$, we denote this operator by

$$\mathcal{H}_v = \hat{\sigma}_v \left( \frac{\oplus_{u \in \text{in}(v)} \mathcal{H}_u}{|\text{in}(v)|} \right) . \tag{8}$$

The reason for using the above notation becomes clear in the sequel. The space $\mathcal{H}_\mathcal{S}$ is obtained by starting with the spaces $\mathcal{H}_v$ corresponding to the input nodes and propagating them according to the structure of $\mathcal{S}$, where at each node $v$ the operation (8) is applied. Hence, to understand $\mathcal{H}_\mathcal{S}$ we need to understand this operation as well as the spaces corresponding to input nodes. The latter spaces are rather simple: for an input node $v$ corresponding to the variable $\mathbf{x}^i$, we have that $\mathcal{H}_v = \{ f_\mathbf{w} \mid \forall \mathbf{x}, \; f_\mathbf{w}(\mathbf{x}) = \langle \mathbf{w}, \mathbf{x}^i \rangle \}$ and $\|f_\mathbf{w}\|_{\mathcal{H}_v} = \|\mathbf{w}\|$. To understand (8), it is convenient to decompose it into two operations. The first operation, termed the *direct average*, is defined through the equation $\tilde{\kappa}_v(\mathbf{x}, \mathbf{x}') = \frac{\sum_{u \in \text{in}(v)} \kappa_u(\mathbf{x}, \mathbf{x}')}{|\text{in}(v)|}$, and the resulting kernel space is denoted $\mathcal{H}_{\tilde{v}} = \frac{\oplus_{u \in \text{in}(v)} \mathcal{H}_u}{|\text{in}(v)|}$. The second operation, called the *extension* according to $\hat{\sigma}_v$, is defined through $\kappa_v(\mathbf{x}, \mathbf{x}') = \hat{\sigma}_v(\tilde{\kappa}_v(\mathbf{x}, \mathbf{x}'))$. The resulting kernel space is denoted $\mathcal{H}_v = \hat{\sigma}_v(\mathcal{H}_{\tilde{v}})$. We next analyze these two operations.

**The direct average of kernel spaces.**  Let $\mathcal{H}_1, \ldots, \mathcal{H}_n$ be kernel spaces with kernels $\kappa_1, \ldots, \kappa_n$ : $\mathcal{X} \times \mathcal{X} \to \mathbb{R}$. Their *direct average*, denoted $\mathcal{H} = \frac{\mathcal{H}_1 \oplus \cdots \oplus \mathcal{H}_n}{n}$, is the kernel space corresponding to the kernel $\kappa(\mathbf{x}, \mathbf{x}') = \frac{1}{n} \sum_{i=1}^n \kappa_i(\mathbf{x}, \mathbf{x}')$.

**Lemma 10.** *The function $\kappa$ is indeed a kernel. Furthermore, the following properties hold.*

1. *If $\mathcal{H}_1, \ldots, \mathcal{H}_n$ are normalized then so is $\mathcal{H}$.*

2. $\mathcal{H} = \left\{ \frac{f_1 + \ldots + f_n}{n} \mid f_i \in \mathcal{H}_i \right\}$

3. $\|f\|_\mathcal{H}^2 = \inf \left\{ \frac{\|f_1\|_{\mathcal{H}_1}^2 + \ldots + \|f_n\|_{\mathcal{H}_n}^2}{n} \;\; s.t. \;\; f = \frac{f_1 + \ldots + f_n}{n}, \; f_i \in \mathcal{H}_i \right\}$

*Proof.* **(outline)** The fact that $\kappa$ is a kernel follows directly from the definition of a kernel and the fact that an average of PSD matrices is PSD. Also, it is straight forward to verify item 1. We now proceed to items 2 and 3. By Theorem 8 there are Hilbert spaces $\mathcal{G}_1, \ldots, \mathcal{G}_n$ and mappings $\Phi_i : \mathcal{X} \to \mathcal{G}_i$ such that $\kappa_i(\mathbf{x}, \mathbf{x}') = \langle \dot\Phi_i(\mathbf{x}), \Phi_i(\mathbf{x}') \rangle_{\mathcal{G}_i}$. Consider now the mapping

$$\Psi(\mathbf{x}) = \left( \frac{\Phi_1(\mathbf{x})}{\sqrt{n}}, \ldots, \frac{\Phi_n(\mathbf{x})}{\sqrt{n}} \right).$$

It holds that $\kappa(\mathbf{x}, \mathbf{x}') = \langle \Psi(\mathbf{x}), \Psi(\mathbf{x}') \rangle$. Properties 2 and 3 now follow directly form Thm. 8 applied to $\Psi$. $\square$

**The extension of a kernel space.** Let $\mathcal{H}$ be a normalized kernel space with a kernel $\kappa$. Let $\mu(x) = \sum_i b_i x^i$ be a PSD function. As we will see shortly, a function is PSD if and only if it is a dual of an activation function. The *extension* of $\mathcal{H}$ w.r.t. $\mu$, denoted $\mu(\mathcal{H})$, is the kernel space corresponding to the kernel $\kappa'(\mathbf{x}, \mathbf{x}') = \mu(\kappa(\mathbf{x}, \mathbf{x}'))$.

**Lemma 11.** *The function $\kappa'$ is indeed a kernel. Furthermore, the following properties hold.*

1. *$\mu(\mathcal{H})$ is normalized if and only if $\mu$ is.*

2. *$\mu(\mathcal{H}) = \overline{\text{span}} \left\{ \prod_{g \in A} g \mid A \subset \mathcal{H}, \ b_{|A|} > 0 \right\}$ where $\overline{\text{span}}(\mathcal{A})$ is the closure of the span of $\mathcal{A}$.*

3. *$\|f\|_{\mu(\mathcal{H})} \leq \inf \left\{ \sum_A \frac{\prod_{g \in A} \|g\|_{\mathcal{H}}}{\sqrt{b_{|A|}}} \ \text{s.t.} \ f = \sum_A \prod_{g \in A} g, \ A \subset \mathcal{H} \right\}$*

*Proof.* **(outline)** Let $\Phi : \mathcal{X} \to \mathcal{G}$ be a mapping from $\mathcal{X}$ to the unit ball of a Hilbert space $\mathcal{G}$ such that $\kappa(\mathbf{x}, \mathbf{x}') = \langle \Phi(\mathbf{x}), \Phi(\mathbf{x}') \rangle$. Define

$$\Psi(\mathbf{x}) = \left( \sqrt{b_0}, \sqrt{b_1}\Phi(\mathbf{x}), \sqrt{b_2}\Phi(\mathbf{x}) \otimes \Phi(\mathbf{x}), \sqrt{b_3}\Phi(\mathbf{x}) \otimes \Phi(\mathbf{x}) \otimes \Phi(\mathbf{x}), \ldots \right)$$

It is not difficult to verify that $\langle \Psi(\mathbf{x}), \Psi(\mathbf{x}') \rangle = \mu(\kappa(\mathbf{x}, \mathbf{x}'))$. Hence, by Thm. 8, $\kappa'$ is indeed a kernel. Verifying property 1 is a straightforward task. Properties 2 and 3 follow by applying Thm. 8 on the mapping $\Psi$. $\square$

## C  The dual activation function

The following lemma describes a few basic properties of the dual activation. These properties follow easily from the definition of the dual activation and equations (2), (4), and (5).

**Lemma 12.** *The following properties of the mapping $\sigma \mapsto \hat\sigma$ hold:*

(a) *If $\sigma = \sum_i a_i h_i$ is the Hermite expansion of $\sigma$, then $\hat\sigma(\rho) = \sum_i a_i^2 \rho^i$.*

(b) *For every $\sigma$, $\hat\sigma$ is positive definite.*

(c) *Every positive definite function is a dual of some activation.*

(d) *The mapping $\sigma \mapsto \hat\sigma$ preserves norms.*

(e) *The mapping $\sigma \mapsto \hat\sigma$ commutes with differentiation.*

(f) *For $a \in \mathbb{R}$, $\widehat{a\sigma} = a^2 \hat\sigma$.*

(g) *For every $\sigma$, $\hat\sigma$ is continuous in $[-1, 1]$ and smooth in $(-1, 1)$.*

(h) *For every $\sigma$, $\hat\sigma$ is non-decreasing and convex in $[0, 1]$.*

(i) *For every $\sigma$, the range of $\hat\sigma$ is $\left[ -\|\sigma\|^2, \|\sigma\|^2 \right]$.*

*(j) For every $\sigma$, $\hat{\sigma}(0) = \left(\mathbb{E}_{X \sim N(0,1)} \sigma(X)\right)^2$ and $\hat{\sigma}(1) = \|\sigma\|^2$.*

We next discuss a few examples for activations and calculate their dual activation and kernel. Note that the dual of the exponential activation was calculated in [29] and the duals of the step and the ReLU activations were calculated in [13]. Here, our derivations are different and may prove useful for future calculations of duals for other activations.

**The exponential activation.** Consider the activation function $\sigma(x) = Ce^{ax}$ where $C > 0$ is a normalization constant such that $\|\sigma\| = 1$. The actual value of $C$ is $e^{-2a^2}$ but it will not be needed for the derivation below. From properties (e) and (f) of Lemma 12 we have that,

$$(\hat{\sigma})' = \widehat{\sigma'} = \widehat{a\sigma} = a^2\hat{\sigma} \, .$$

The the solution of ordinary differential equation $(\hat{\sigma})' = a^2\hat{\sigma}$ is of the form $\hat{\sigma}(\rho) = b \exp\left(a^2\rho\right)$. Since $\hat{\sigma}(1) = 1$ we have $b = e^{-a^2}$. We therefore obtain that the dual activation function is

$$\hat{\sigma}(\rho) = e^{a^2\rho - a^2} = e^{a^2(\rho-1)} \, .$$

Note that the kernel induced by $\sigma$ is the RBF kernel, restricted to the $d$-dimensional sphere,

$$\kappa_\sigma(\mathbf{x}, \mathbf{x}') = e^{a^2(\langle \mathbf{x}, \mathbf{x}' \rangle - 1)} = e^{-\frac{a^2\|\mathbf{x} - \mathbf{x}'\|^2}{2}} \, .$$

**The Sine activation and the Sinh kernel.** Consider the activation $\sigma(x) = \sin(ax)$. We can write $\sin(ax) = \frac{e^{iax} - e^{-iax}}{2i}$. We have

$$
\begin{aligned}
\hat{\sigma}(\rho) &= \mathbb{E}_{(X,Y) \sim N_\rho} \left( \frac{e^{iaX} - e^{-iaX}}{2i} \right) \left( \frac{e^{iaY} - e^{-iaY}}{2i} \right) \\
&= -\frac{1}{4} \mathbb{E}_{(X,Y) \sim N_\rho} \left( e^{iaX} - e^{-iaX} \right) \left( e^{iaY} - e^{-iaY} \right) \\
&= -\frac{1}{4} \mathbb{E}_{(X,Y) \sim N_\rho} \left[ e^{ia(X+Y)} - e^{ia(X-Y)} - e^{ia(-X+Y)} + e^{ia(-X-Y)} \right] \, .
\end{aligned}
$$

Recall that the characteristic function, $\mathbb{E}[e^{itX}]$, when $X$ is distributed $N(0,1)$ is $e^{-\frac{1}{2}t^2}$. Since $X + Y$ and $-X - Y$ are normal variables with expectation 0 and variance of $2 + 2\rho$, it follows that,

$$\mathbb{E}_{(X,Y) \sim N_\rho} e^{ia(X+Y)} = \mathbb{E}_{(X,Y) \sim N_\rho} e^{-ia(X+Y)} = e^{-\frac{a^2(2+2\rho)}{2}} \, .$$

Similarly, since the variance of $X - Y$ and $Y - X$ is $2 - 2\rho$, we get

$$\mathbb{E}_{(X,Y) \sim N_\rho} e^{ia(X-Y)} = \mathbb{E}_{(X,Y) \sim N_\rho} e^{ia(-X+Y)} = e^{-\frac{a^2(2-2\rho)}{2}} \, .$$

We therefore obtain that

$$\hat{\sigma}(\rho) = \frac{e^{-a^2(1-\rho)} - e^{-a^2(1+\rho)}}{2} = e^{-a^2} \sinh(a^2\rho) \, .$$

**Hermite activations and polynomial kernels.** From Lemma 12 it follows that the dual activation of the Hermite polynomial $h_n$ is $\hat{h}_n(\rho) = \rho^n$. Hence, the corresponding kernel is the polynomial kernel.

**The normalized step activation.** Consider the activation

$$\sigma(x) = \begin{cases} \sqrt{2} & x > 0 \\ 0 & x \leq 0 \end{cases} \, .$$

To calculate $\hat{\sigma}$ we compute the Hermite expansion of $\sigma$. For $n \geq 0$ we let

$$a_n = \frac{1}{\sqrt{2\pi}} \int_{-\infty}^{\infty} \sigma(x) h_n(x) e^{-\frac{x^2}{2}} dx = \frac{1}{\sqrt{\pi}} \int_0^{\infty} h_n(x) e^{-\frac{x^2}{2}} dx \, .$$

Since $h_0(x) = 1$, $h_1(x) = x$, and $h_2(x) = \frac{x^2-1}{\sqrt{2}}$, we get the corresponding coefficients,

$$a_0 = \mathop{\mathbb{E}}_{X \sim N(0,1)} [\sigma(X)] = \frac{1}{\sqrt{2}}$$

$$a_1 = \mathop{\mathbb{E}}_{X \sim N(0,1)} [\sigma(X)X] = \frac{1}{\sqrt{2}} \mathop{\mathbb{E}}_{X \sim N(0,1)} [|X|] = \frac{1}{\sqrt{\pi}}$$

$$a_2 = \frac{1}{\sqrt{2}} \mathop{\mathbb{E}}_{X \sim N(0,1)} [\sigma(X)(X^2 - 1)] = \frac{1}{2} \mathop{\mathbb{E}}_{X \sim N(0,1)} [X^2 - 1] = 0 \,.$$

For $n \geq 3$ we write $g_n(x) = h_n(x)e^{-\frac{x^2}{2}}$ and note that

$$\begin{aligned}
g_n'(x) &= \left[ h_n'(x) - xh_n(x) \right] e^{-\frac{x^2}{2}} \\
&= \left[ \sqrt{n}h_{n-1}(x) - xh_n(x) \right] e^{-\frac{x^2}{2}} \\
&= -\sqrt{n+1}\, h_{n+1}(x)e^{-\frac{x^2}{2}} \\
&= -\sqrt{n+1}\, g_{n+1}(x) \,.
\end{aligned}$$

Here, the second equality follows from (4) and the third form (3). We therefore get

$$\begin{aligned}
a_n &= \frac{1}{\sqrt{\pi}} \int_0^\infty g_n(x)dx \\
&= -\frac{1}{\sqrt{n\pi}} \int_0^\infty g_{n-1}'(x)dx \\
&= \frac{1}{\sqrt{n\pi}} \left( g_{n-1}(0) - \overbrace{g_{n-1}(\infty)}^{=0} \right) \\
&= \frac{1}{\sqrt{n\pi}} h_{n-1}(0) \\
&= \begin{cases} \frac{(-1)^{\frac{n-1}{2}}(n-2)!!}{\sqrt{n\pi}\sqrt{(n-1)!}} = \frac{(-1)^{\frac{n-1}{2}}(n-2)!!}{\sqrt{\pi n!}} & \text{if } n \text{ is odd} \\ 0 & \text{if } n \text{ is even} \end{cases} \,.
\end{aligned}$$

The second equality follows from (3) and the last equality follows from (6). Finally, from Lemma 12 we have that $\hat{\sigma}(\rho) = \sum_{n=0}^\infty b_n \rho^n$ where

$$b_n = \begin{cases} \frac{((n-2)!!)^2}{\pi n!} & \text{if } n \text{ is odd} \\ \frac{1}{2} & \text{if } n = 0 \\ 0 & \text{if } n \text{ is even} \geq 2 \end{cases} \,.$$

In particular, $(b_0, b_1, b_2, b_3, b_4, b_5, b_6) = \left( \frac{1}{2}, \frac{1}{\pi}, 0, \frac{1}{6\pi}, 0, \frac{3}{40\pi}, 0 \right)$. Note that from the Taylor expansion of $\cos^{-1}$ it follows that $\hat{\sigma}(\rho) = 1 - \frac{\cos^{-1}(\rho)}{\pi}$.

**The normalized ReLU activation.** Consider the activation $\sigma(x) = \sqrt{2}\max(0, x)$. We now write $\hat{\sigma}(\rho) = \sum_i b_i \rho^i$. The first coefficient is

$$b_0 = \left( \mathop{\mathbb{E}}_{X \sim N(0,1)} \sigma(X) \right)^2 = \frac{1}{2} \left( \mathop{\mathbb{E}}_{X \sim N(0,1)} |X| \right)^2 = \frac{1}{\pi} \,.$$

To calculate the remaining coefficients we simply note that the derivative of the ReLU activation is the step activation and the mapping $\sigma \mapsto \hat{\sigma}$ commutes with differentiation. Hence, from the calculation of the step activation we get,

$$b_n = \begin{cases} \frac{((n-3)!!)^2}{\pi n!} & \text{if } n \text{ is even} \\ \frac{1}{2} & \text{if } n = 1 \\ 0 & \text{if } n \text{ is odd} \geq 3 \end{cases} \,.$$

In particular, $(b_0, b_1, b_2, b_3, b_4, b_5, b_6) = \left( \frac{1}{\pi}, \frac{1}{2}, \frac{1}{2\pi}, 0, \frac{1}{24\pi}, 0, \frac{1}{80\pi} \right)$. We see that the coefficients corresponding to the degrees 0, 1, and 2 sum to 0.9774. The sums up to degrees 4 or 6 are 0.9907 and 0.9947 respectively. That is, we get an excellent approximation of less than 1% error with a dual activation of degree 4.

**The collapsing tower of fully connected layers.**    To conclude this section we discuss the case of very deep networks. The setting is taken for illustrative purposes but it might surface when building networks with numerous fully connected layers. Indeed, most deep architectures that we are aware of do not employ more than five *consecutive* fully connected layers.

Consider a skeleton $\mathcal{S}_m$ consisting of $m$ fully connected layers, each layer associated with the same (normalized) activation $\sigma$. We would like to examine the form of the compositional kernel as the number of layers becomes very large. Due to the repeated structure and activation we have

$$\kappa_{\mathcal{S}_m}(\mathbf{x}, \mathbf{y}) = \alpha_m \left( \frac{\langle \mathbf{x}, \mathbf{y} \rangle}{n} \right) \quad \text{where} \quad \alpha_m = \hat{\sigma}^m = \overbrace{\hat{\sigma} \circ \ldots \circ \hat{\sigma}}^{m \text{ times}} .$$

Hence, the limiting properties of $\kappa_{\mathcal{S}_m}$ can be understood from the limit of $\alpha_m$. In the case that $\sigma(x) = x$ or $\sigma(x) = -x$, $\hat{\sigma}$ is the identity function. Therefore $\alpha_m(\rho) = \hat{\sigma}(\rho) = \rho$ for all $m$ and $\kappa_{\mathcal{S}_m}$ is simply the linear kernel. Assume now that $\sigma$ is neither the identity nor its negation. The following claim shows that $\alpha_m$ has a point-wise limit corresponding to a degenerate kernel.

**Claim 1.** *There exists a constant* $0 \leq \alpha_\sigma \leq 1$ *such that for all* $-1 < \rho < 1$,

$$\lim_{m \to \infty} \alpha_m(\rho) = \alpha_\sigma$$

Before proving the claim, we note that for $\rho = 1$, $\alpha_m(1) = 1$ for all $m$, and therefore $\lim_{m \to \infty} \alpha_m(1) = 1$. For $\rho = -1$, if $\sigma$ is anti-symmetric then $\alpha_m(-1) = -1$ for all $m$, and in particular $\lim_{m \to \infty} \alpha_m(-1) = -1$. In any other case, our argument can show that $\lim_{m \to \infty} \alpha_m(-1) = \alpha_\sigma$.

*Proof.* Recall that $\hat{\sigma}(\rho) = \sum_{i=0}^{\infty} b_i \rho^i$ where the $b_i$'s are non-negative numbers that sum to 1. By the assumption that $\sigma$ is not the identity or its negation, $b_1 < 1$. We first claim that there is a unique $\alpha_\sigma \in [0, 1]$ such that

$$\forall x \in (-1, \alpha_\sigma), \; \hat{\sigma}(\rho) > \rho \text{ and } \; \forall x \in (\alpha_\sigma, 1), \; \alpha_\sigma < \hat{\sigma}(\rho) < \rho \tag{9}$$

To prove (9) it suffices to prove the following properties.

   (a)  $\hat{\sigma}(\rho) > \rho$ for $\rho \in (-1, 0)$

   (b)  $\hat{\sigma}$ is non-decreasing and convex in $[0, 1]$

   (c)  $\hat{\sigma}(1) = 1$

   (d)  the graph of $\hat{\sigma}$ has at most a single intersection in $[0, 1)$ with the graph of $f(\rho) = \rho$

If the above properties hold we can take $\alpha_\sigma$ to be the intersection point or 1 if such a point does not exist. We first show (a). For $\rho \in (-1, 0)$ we have that

$$\hat{\sigma}(\rho) \;=\; b_0 + \sum_{i=1}^{\infty} b_i \rho^i \;\geq\; b_0 - \sum_{i=1}^{\infty} b_i |\rho|^i \;>\; - \sum_{i=1}^{\infty} b_i |\rho| \;\geq\; -|\rho| \;=\; \rho \,.$$

Here, the third inequality follows form the fact that $b_0 \geq 0$ and for all $i$, $-b_i |\rho|^i \geq -b_i |\rho|$. Moreover since $b_1 < 1$, one of these inequalities must be strict. Properties (b) and (c) follows from Lemma 12. Finally, to show (d), we note that the second derivative of $\hat{\sigma}(\rho) - \rho$ is $\sum_{i \geq 2} i(i-1) b_i \rho^{i-2}$ which is non-negative in $[0, 1)$. Hence, $\hat{\sigma}(\rho) - \rho$ is convex in $[0, 1]$ and in particular intersects with the $x$-axis at either 0, 1, 2 or infinitely many times in $[0, 1]$. As we assume that $\hat{\sigma}$ is not the identity, we can rule out the option of infinitely many intersections. Also, since $\hat{\sigma}(1) = 1$, we know that there is at least one intersection in $[0, 1]$. Hence, there are 1 or 2 intersections in $[0, 1]$ and because one of them is in $\rho = 1$, we conclude that there is at most one intersection in $[0, 1)$.

Lastly, we derive the conclusion of the claim from equation (9). Fix $\rho \in (-1, 1)$. Assume first that $\rho \geq \alpha_\sigma$. By (9), $\alpha_m(\rho)$ is a monotonically non-increasing sequence that is lower bounded by $\alpha_\sigma$. Hence, it has a limit $\alpha_\sigma \leq \tau \leq \rho < 1$. Now, by the continuity of $\hat{\sigma}$ we have

$$\hat{\sigma}(\tau) = \hat{\sigma} \left( \lim_{m \to \infty} \alpha_m(\rho) \right) = \lim_{m \to \infty} \hat{\sigma}(\alpha_m(\rho)) = \lim_{m \to \infty} \alpha_{m+1}(\rho) = \tau \,.$$

Since the only solution to $\hat{\sigma}(\rho) = \rho$ in $(-1, 1)$ is $\alpha_\sigma$, we must have $\tau = \alpha_\sigma$. We next deal with the case that $-1 < \rho < \alpha_\sigma$. If for some $m$, $\alpha_m(\rho) \in [\alpha_\sigma, 1)$, the argument for $\alpha_\sigma \leq \rho$ shows that $\alpha_\sigma = \lim_{m \to \infty} \alpha_m(\rho)$. If this is not the case, we have that for all $m$, $\alpha_m(\rho) \leq \alpha_{m+1}(\rho) \leq \alpha_\sigma$. As in the case of $\rho \geq \alpha_\sigma$, this can be used to show that $\alpha_m(\rho)$ converges to $\alpha_\sigma$. $\qquad\square$

# D Proofs

## D.1 Well-behaved activations

The proof of our main results applies to activations that are decent, i.e. well-behaved, in a sense defined in the sequel. We then show that $C$-bounded activations as well as the ReLU activation are decent. We first need to extend the definition of the dual activation and kernel to apply to vectors in $\mathbb{R}^d$, rather than just $\mathbb{S}^d$. We denote by $\mathcal{M}_+$ the collection of $2 \times 2$ positive semi-define matrices and by $\mathcal{M}_{++}$ the collection of positive definite matrices.

**Definition.** *Let $\sigma$ be an activation. Define the following,*

$$\bar{\sigma} : \mathcal{M}_+^2 \to \mathbb{R} \ , \quad \bar{\sigma}(\Sigma) = \mathop{\mathbb{E}}_{(X,Y) \sim N(0,\Sigma)} \sigma(X)\sigma(Y) \ , \quad k_\sigma(\mathbf{x}, \mathbf{y}) = \bar{\sigma} \begin{pmatrix} \|\mathbf{x}\|^2 & \langle \mathbf{x}, \mathbf{y} \rangle \\ \langle \mathbf{x}, \mathbf{y} \rangle & \|\mathbf{y}\|^2 \end{pmatrix} .$$

We underscore the following properties of the extension of a dual activation.

    (a) The following equality holds,

$$\hat{\sigma}(\rho) = \bar{\sigma} \begin{pmatrix} 1 & \rho \\ \rho & 1 \end{pmatrix}$$

    (b) The restriction of the extended $k_\sigma$ to the sphere agrees with the restricted definition.

    (c) The extended dual activation and kernel are defined for every activation $\sigma$ such that for all $a \geq 0$, $x \mapsto \sigma(ax)$ is square integrable with respect to the Gaussian measure.

    (d) For $\mathbf{x}, \mathbf{y} \in \mathbb{R}^d$, if $\mathbf{w} \in \mathbb{R}^d$ is a multivariate normal distribution with zero mean vector and identity covariance matrix, then

$$k_\sigma(\mathbf{x}, \mathbf{y}) = \mathop{\mathbb{E}}_{\mathbf{w}} \sigma(\langle \mathbf{w}, \mathbf{x} \rangle) \sigma(\langle \mathbf{w}, \mathbf{y} \rangle) .$$

Denote

$$\mathcal{M}_+^\gamma := \left\{ \begin{pmatrix} \Sigma_{11} & \Sigma_{12} \\ \Sigma_{12} & \Sigma_{22} \end{pmatrix} \in \mathcal{M}_+ \mid 1 - \gamma \leq \Sigma_{11}, \Sigma_{22} \leq 1 + \gamma \right\} .$$

**Definition.** *A normalized activation $\sigma$ is $(\alpha, \beta, \gamma)$-decent for $\alpha, \beta, \gamma \geq 0$ if the following conditions hold.*

    (i) *The dual activation $\bar{\sigma}$ is $\beta$-Lipschitz in $\mathcal{M}_+^\gamma$ with respect to the $\infty$-norm.*

    (ii) *If $(X_1, Y_1), \ldots, (X_r, Y_r)$ are independent samples from $N(0, \Sigma)$ for $\Sigma \in \mathcal{M}_+^\gamma$ then*

$$\Pr\left( \left| \frac{\sum_{i=1}^r \sigma(X_i)\sigma(Y_i)}{r} - \bar{\sigma}(\Sigma) \right| \geq \epsilon \right) \leq 2 \exp\left( -\frac{r\epsilon^2}{2\alpha^2} \right) .$$

**Lemma 13** (Bounded activations are decent). *Let $\sigma : \mathbb{R} \to \mathbb{R}$ be a $C$-bounded normalized activation. Then, $\sigma$ is $(C^2, 2C^2, \gamma)$-decent for all $\gamma \geq 0$.*

*Proof.* It is enough to show that the following properties hold.

    1. The (extended) dual activation $\bar{\sigma}$ is $2C^2$-Lipschitz in $\mathcal{M}_{++}$ w.r.t. the $\infty$-norm.

    2. If $(X_1, Y_1), \ldots, (X_r, Y_r)$ are independent samples from $N(0, \Sigma)$ then

$$\Pr\left( \left| \frac{\sum_{i=1}^r \sigma(X_i)\sigma(Y_i)}{r} - \bar{\sigma}(\Sigma) \right| \geq \epsilon \right) \leq 2 \exp\left( -\frac{r\epsilon^2}{2C^4} \right)$$

From the boundedness of $\sigma$ it holds that $|\sigma(X)\sigma(Y)| \le C^2$. Hence, the second property follows directly from Hoeffding's bound. We next prove the first part. Let $\mathbf{z} = (x, y)$ and $\phi(\mathbf{z}) = \sigma(x)\sigma(y)$. Note that for $\Sigma \in \mathcal{M}_{++}$ we have

$$\bar{\sigma}(\Sigma) = \frac{1}{2\pi\sqrt{\det(\Sigma)}} \int_{\mathbb{R}^2} \phi(\mathbf{z}) e^{-\frac{\mathbf{z}^\top \Sigma^{-1} \mathbf{z}}{2}} d\mathbf{z}.$$

Thus we get that,

$$
\begin{aligned}
\frac{\partial \bar{\sigma}}{\partial \Sigma} &= \frac{1}{2\pi} \int_{\mathbb{R}^2} \phi(\mathbf{z}) \left[ \frac{\frac{1}{2}\sqrt{\det(\Sigma)}\Sigma^{-1} - \frac{1}{2}\sqrt{\det(\Sigma)}(\Sigma^{-1}\mathbf{z}\mathbf{z}^\top\Sigma^{-1})}{\det(\Sigma)} \right] e^{-\frac{\mathbf{z}^\top \Sigma^{-1} \mathbf{z}}{2}} d\mathbf{z} \\
&= \frac{1}{2\pi\sqrt{\det(\Sigma)}} \int_{\mathbb{R}^2} \phi(\mathbf{z}) \frac{1}{2} \left[ \Sigma^{-1} - \Sigma^{-1}\mathbf{z}\mathbf{z}^\top\Sigma^{-1} \right] e^{-\frac{\mathbf{z}^\top \Sigma^{-1} \mathbf{z}}{2}} d\mathbf{z}
\end{aligned}
$$

Let $g(\mathbf{z}) = e^{-\frac{\mathbf{z}^\top \Sigma^{-1} \mathbf{z}}{2}}$. Then, the first and second order partial derivatives of $g$ are

$$
\begin{aligned}
\frac{\partial g}{\partial \mathbf{z}} &= -\Sigma^{-1}\mathbf{z}e^{-\frac{\mathbf{z}^\top \Sigma^{-1} \mathbf{z}}{2}} \\
\frac{\partial^2 g}{\partial^2 \mathbf{z}} &= \left[ -\Sigma^{-1} + \Sigma^{-1}\mathbf{z}\mathbf{z}^\top\Sigma^{-1} \right] e^{-\frac{\mathbf{z}^\top \Sigma^{-1} \mathbf{z}}{2}}.
\end{aligned}
$$

We therefore obtain that,

$$\frac{\partial \bar{\sigma}}{\partial \Sigma} = -\frac{1}{4\pi\sqrt{\det(\Sigma)}} \int_{\mathbb{R}^2} \phi \frac{\partial^2 g}{\partial^2 \mathbf{z}} d\mathbf{z}.$$

By the product rule we have

$$\frac{\partial \bar{\sigma}}{\partial \Sigma} = -\frac{1}{2\pi\sqrt{\det(\Sigma)}} \frac{1}{2} \int_{\mathbb{R}^2} \frac{\partial^2 \phi}{\partial^2 \mathbf{z}} g d\mathbf{z} = -\frac{1}{2} \mathop{\mathbb{E}}_{(X,Y)\sim\mathrm{N}(0,\Sigma)} \left[ \frac{\partial^2 \phi}{\partial^2 \mathbf{z}}(X, Y) \right]$$

We conclude that $\bar{\sigma}$ is differentiable in $\mathcal{M}_{++}$ with partial derivatives that are point-wise bounded by $\frac{C^2}{2}$. Thus, $\bar{\sigma}$ is $2C^2$-Lipschitz in $\mathcal{M}_+$ w.r.t. the $\infty$-norm. $\qquad\square$

We next show that the ReLU activation is decent.

**Lemma 14** (ReLU is decent)**.** *There exists a constant $\alpha_{\mathrm{ReLU}} \ge 1$ such that for $0 \le \gamma \le 1$, the normalized ReLU activation $\sigma(x) = \sqrt{2}\max(0, x)$ is $(\alpha_{\mathrm{ReLU}}, 1 + o(\gamma), \gamma)$-decent.*

*Proof.* The measure concentration property follows from standard concentration bounds for sub-exponential random variables (e.g. [43]). It remains to show that $\bar{\sigma}$ is $(1 + o(\gamma))$-Lipschitz in $\mathcal{M}_+^\gamma$. We first calculate an exact expression for $\bar{\sigma}$. The expression was already calculated in [13], yet we give here a derivation for completeness.

**Claim 2.** *The following equality holds for all $\Sigma \in \mathcal{M}_+^2$,*

$$\bar{\sigma}(\Sigma) = \sqrt{\Sigma_{11}\Sigma_{22}}\,\hat{\sigma}\left( \frac{\Sigma_{12}}{\sqrt{\Sigma_{11}\Sigma_{22}}} \right).$$

*Proof.* Let us denote

$$\tilde{\Sigma} = \begin{pmatrix} 1 & \frac{\Sigma_{12}}{\sqrt{\Sigma_{11}\Sigma_{12}}} \\ \frac{\Sigma_{12}}{\sqrt{\Sigma_{11}\Sigma_{12}}} & 1 \end{pmatrix}.$$

By the positive homogeneity of the ReLU activation we have

$$
\begin{aligned}
\bar{\sigma}(\Sigma) &= \mathop{\mathbb{E}}_{(X,Y)\sim\mathrm{N}(0,\Sigma)} \sigma(X)\sigma(Y) \\
&= \sqrt{\Sigma_{11}\Sigma_{22}} \mathop{\mathbb{E}}_{(X,Y)\sim\mathrm{N}(0,\Sigma)} \sigma\left( \frac{X}{\sqrt{\Sigma_{11}}} \right)\sigma\left( \frac{Y}{\sqrt{\Sigma_{22}}} \right) \\
&= \sqrt{\Sigma_{11}\Sigma_{22}} \mathop{\mathbb{E}}_{(\tilde{X},\tilde{Y})\sim\mathrm{N}(0,\tilde{\Sigma})} \sigma\left( \tilde{X} \right)\sigma\left( \tilde{Y} \right) \\
&= \sqrt{\Sigma_{11}\Sigma_{22}}\,\hat{\sigma}\left( \frac{\Sigma_{12}}{\sqrt{\Sigma_{11}\Sigma_{22}}} \right).
\end{aligned}
$$

which concludes the proof. $\qquad\square$

For brevity, we henceforth drop the argument from $\bar{\sigma}(\Sigma)$ and use the abbreviation $\bar{\sigma}$. In order to show that $\bar{\sigma}$ is $(1 + o(\gamma))$-Lipschitz w.r.t. the $\infty$-norm it is enough to show that for every $\Sigma \in \mathcal{M}_+^\gamma$ we have,

$$\|\nabla\bar{\sigma}\|_1 = \left|\frac{\partial\bar{\sigma}}{\partial\Sigma_{12}}\right| + \left|\frac{\partial\bar{\sigma}}{\partial\Sigma_{11}}\right| + \left|\frac{\partial\bar{\sigma}}{\partial\Sigma_{22}}\right| \leq 1 + o(\gamma) . \tag{10}$$

First, Note that $\partial\bar{\sigma}/\partial\Sigma_{11}$ and $\partial\bar{\sigma}/\partial\Sigma_{22}$ have the same sign, hence,

$$\|\nabla\bar{\sigma}\|_1 = \left|\frac{\partial\bar{\sigma}}{\partial\Sigma_{12}}\right| + \left|\frac{\partial\bar{\sigma}}{\partial\Sigma_{11}} + \frac{\partial\bar{\sigma}}{\partial\Sigma_{22}}\right| .$$

Next we get that,

$$\begin{aligned}
\frac{\partial\bar{\sigma}}{\partial\Sigma_{11}} &= \frac{1}{2}\sqrt{\frac{\Sigma_{22}}{\Sigma_{11}}}\,\hat{\sigma}\left(\frac{\Sigma_{12}}{\sqrt{\Sigma_{11}\Sigma_{22}}}\right) - \frac{1}{2}\sqrt{\frac{\Sigma_{22}}{\Sigma_{11}}}\frac{\Sigma_{12}}{\sqrt{\Sigma_{11}\Sigma_{22}}}\,\hat{\sigma}'\left(\frac{\Sigma_{12}}{\sqrt{\Sigma_{11}\Sigma_{22}}}\right) \\
\frac{\partial\bar{\sigma}}{\partial\Sigma_{22}} &= \frac{1}{2}\sqrt{\frac{\Sigma_{11}}{\Sigma_{22}}}\,\hat{\sigma}\left(\frac{\Sigma_{12}}{\sqrt{\Sigma_{11}\Sigma_{22}}}\right) - \frac{1}{2}\sqrt{\frac{\Sigma_{11}}{\Sigma_{22}}}\frac{\Sigma_{12}}{\sqrt{\Sigma_{11}\Sigma_{22}}}\,\hat{\sigma}'\left(\frac{\Sigma_{12}}{\sqrt{\Sigma_{11}\Sigma_{22}}}\right) \\
\frac{\partial\bar{\sigma}}{\partial\Sigma_{12}} &= \hat{\sigma}'\left(\frac{\Sigma_{12}}{\sqrt{\Sigma_{11}\Sigma_{22}}}\right) .
\end{aligned}$$

We therefore get that the 1-norm of $\nabla\bar{\sigma}$ is,

$$\|\nabla\bar{\sigma}\|_1 = \frac{1}{2}\frac{\Sigma_{11} + \Sigma_{22}}{\sqrt{\Sigma_{11}\Sigma_{22}}}\left|\hat{\sigma}\left(\frac{\Sigma_{12}}{\sqrt{\Sigma_{11}\Sigma_{22}}}\right) - \frac{\Sigma_{12}}{\sqrt{\Sigma_{11}\Sigma_{22}}}\,\hat{\sigma}'\left(\frac{\Sigma_{12}}{\sqrt{\Sigma_{11}\Sigma_{22}}}\right)\right| + \hat{\sigma}'\left(\frac{\Sigma_{12}}{\sqrt{\Sigma_{11}\Sigma_{22}}}\right) .$$

The gradient of $\frac{1}{2}\frac{\Sigma_{11}+\Sigma_{22}}{\sqrt{\Sigma_{11}\Sigma_{22}}}$ at $(\Sigma_{11}, \Sigma_{22}) = (1, 1)$ is $(0, 0)$. Therefore, from the mean value theorem we get, $\frac{1}{2}\frac{\Sigma_{11}+\Sigma_{22}}{\sqrt{\Sigma_{11}\Sigma_{22}}} = 1 + o(\gamma)$. Furthermore, $\hat{\sigma}$, $\hat{\sigma}'$ and $\frac{\Sigma_{12}}{\sqrt{\Sigma_{11}\Sigma_{22}}}$ are bounded by 1 in absolute value. Hence, we can write,

$$\|\nabla\bar{\sigma}\|_1 = \left|\hat{\sigma}\left(\frac{\Sigma_{12}}{\sqrt{\Sigma_{11}\Sigma_{22}}}\right) - \frac{\Sigma_{12}}{\sqrt{\Sigma_{11}\Sigma_{22}}}\hat{\sigma}'\left(\frac{\Sigma_{12}}{\sqrt{\Sigma_{11}\Sigma_{22}}}\right)\right| + \hat{\sigma}'\left(\frac{\Sigma_{12}}{\sqrt{\Sigma_{11}\Sigma_{22}}}\right) + o(\gamma) .$$

Finally, if we let $t = \frac{\Sigma_{12}}{\sqrt{\Sigma_{11}\Sigma_{22}}}$, we can further simply the expression for $\nabla\bar{\sigma}$,

$$\begin{aligned}
\|\nabla\bar{\sigma}(\Sigma)\|_1 &= |\hat{\sigma}(t) - t\hat{\sigma}'(t)| + |\hat{\sigma}'(t)| + o(\gamma) \\
&= \frac{\sqrt{1-t^2}}{\pi} + 1 - \frac{\cos^{-1}(t)}{\pi} + o(\gamma) .
\end{aligned}$$

Finally, the proof is obtained from the fact that the function $f(t) = \frac{\sqrt{1-t^2}}{\pi} + 1 - \frac{\cos^{-1}(t)}{\pi}$ satisfies $0 \leq f(t) \leq 1$ for every $t \in [-1, 1]$. Indeed, it is simple to verify that $f(-1) = 0$ and $f(1) = 1$. Hence, it suffices to show that $f'$ is non-negative in $[-1, 1]$ which is indeed the case since,

$$f'(t) = \frac{1}{\pi}\frac{1-t}{\sqrt{1-t^2}} = \frac{1}{\pi}\sqrt{\frac{1-t}{1+t}} \geq 0 . \qquad\square$$

### D.2 Proofs of Thms. 3 and 4

We start by an additional theorem which serves as a simple stepping stone for proving the aforementioned main theorems.

**Theorem 15.** *Let $\mathcal{S}$ be a skeleton with $(\alpha, \beta, \gamma)$-decent activations, $0 < \epsilon \leq \gamma$, and $B_d = \sum_{i=0}^{d-1}\beta^i$. Let $\mathbf{w}$ be a random initialization of the network $\mathcal{N} = \mathcal{N}(\mathcal{S}, r)$ with*

$$r \geq \frac{2\alpha^2 B_{\mathrm{depth}(\mathcal{S})}^2 \log\left(\frac{8|\mathcal{S}|}{\delta}\right)}{\epsilon^2} .$$

*Then, for every $\mathbf{x}, \mathbf{y}$ with probability of at least $1 - \delta$, it holds that*

$$|\kappa_{\mathbf{w}}(\mathbf{x}, \mathbf{y}) - \kappa_{\mathcal{S}}(\mathbf{x}, \mathbf{y})| \leq \epsilon .$$

Before proving the theorem we show that together with Lemmas 13 and 14, Theorems 3 and 4 follow from Theorem 15. We restate them as corollaries, prove them, and then proceed to the proof of Theorem 15.

**Corollary 16.** *Let $\mathcal{S}$ be a skeleton with $C$-bounded activations. Let $\mathbf{w}$ be a random initialization of $\mathcal{N} = \mathcal{N}(\mathcal{S}, r)$ with*

$$r \geq \frac{(4C^4)^{\text{depth}(\mathcal{S})+1} \log\left(\frac{8|\mathcal{S}|}{\delta}\right)}{\epsilon^2}.$$

*Then, for every $\mathbf{x}, \mathbf{y}$, w.p. $\geq 1 - \delta$,*

$$|\kappa_{\mathbf{w}}(\mathbf{x}, \mathbf{y}) - \kappa_{\mathcal{S}}(\mathbf{x}, \mathbf{y})| \leq \epsilon.$$

*Proof.* From Lemma 13, for all $\gamma > 0$, each activation is $(C^2, 2C^2, \gamma)$-decent. By Theorem 15, it suffices to show that

$$2\left(C^2\right)^2 \left(\sum_{i=0}^{\text{depth}(\mathcal{S})-1} (2C^2)^i\right)^2 \leq (4C^4)^{\text{depth}(\mathcal{S})+1}.$$

The sum of can be bounded above by,

$$\sum_{i=0}^{\text{depth}(\mathcal{S})-1} (2C^2)^i = \frac{(2C^2)^{\text{depth}(\mathcal{S})} - 1}{2C^2 - 1} \leq \frac{(2C^2)^{\text{depth}(\mathcal{S})}}{C^2}.$$

Therefore, we get that,

$$2\left(C^2\right)^2 \left(\sum_{i=0}^{\text{depth}(\mathcal{S})-1} (2C^2)^i\right)^2 \leq \frac{2C^4(4C^4)^{\text{depth}(\mathcal{S})}}{C^4} \leq (4C^4)^{\text{depth}(\mathcal{S})+1},$$

which concludes the proof. $\qquad\square$

**Corollary 17.** *Let $\mathcal{S}$ be a skeleton with ReLU activations, and $\mathbf{w}$ a random initialization of $\mathcal{N}(\mathcal{S}, r)$ with $r \geq c_1 \frac{\text{depth}^2(\mathcal{S}) \log\left(\frac{8|\mathcal{S}|}{\delta}\right)}{\epsilon^2}$. For all $\mathbf{x}, \mathbf{y}$ and $\epsilon \leq \min(c_2, \frac{1}{\text{depth}(\mathcal{S})})$, w.p. $\geq 1 - \delta$,*

$$|\kappa_{\mathbf{w}}(\mathbf{x}, \mathbf{y}) - \kappa_{\mathcal{S}}(\mathbf{x}, \mathbf{y})| \leq \epsilon$$

*Here, $c_1, c_2 > 0$ are universal constants.*

*Proof.* From Lemma 14, each activation is $(\alpha_{\text{ReLU}}, 1 + o(\epsilon), \epsilon)$-decent. By Theorem 15, it is enough to show that

$$\sum_{i=0}^{\text{depth}(\mathcal{S})-1} (1 + o(\epsilon))^i = O(\text{depth}(\mathcal{S})).$$

This claim follows from the fact that $(1 + o(\epsilon))^i \leq e^{o(\epsilon)\text{depth}(\mathcal{S})}$ as long as $i \leq \text{depth}(\mathcal{S})$. Since we assume that $\epsilon \leq 1/\text{depth}(\mathcal{S})$, the expression is bounded by $e$ for sufficiently small $\epsilon$. $\qquad\square$

We next prove Theorem 15.

*Proof.* (Theorem 15) For a node $u \in \mathcal{S}$ we denote by $\Psi_{u,\mathbf{w}} : \mathcal{X} \to \mathbb{R}^r$ the normalized representation of $\mathcal{S}$'s sub-skeleton rooted at $u$. Analogously, $\kappa_{u,\mathbf{w}}$ denotes the empirical kernel of that network. When $u$ is the output node of $\mathcal{S}$ we still use $\Psi_{\mathbf{w}}$ and $\kappa_{\mathbf{w}}$ for $\Psi_{u,\mathbf{w}}$ and $\kappa_{u,\mathbf{w}}$. Given two fixed $\mathbf{x}, \mathbf{y} \in \mathcal{X}$ and a node $u \in \mathcal{S}$, we denote

$$\mathcal{K}^u_{\mathbf{w}} = \begin{pmatrix} \kappa_{u,\mathbf{w}}(\mathbf{x}, \mathbf{x}) & \kappa_{u,\mathbf{w}}(\mathbf{x}, \mathbf{y}) \\ \kappa_{u,\mathbf{w}}(\mathbf{x}, \mathbf{y}) & \kappa_{u,\mathbf{w}}(\mathbf{y}, \mathbf{y}) \end{pmatrix}, \quad \mathcal{K}^u = \begin{pmatrix} \kappa_u(\mathbf{x}, \mathbf{x}) & \kappa_u(\mathbf{x}, \mathbf{y}) \\ \kappa_u(\mathbf{x}, \mathbf{y}) & \kappa_u(\mathbf{y}, \mathbf{y}) \end{pmatrix}$$

$$\mathcal{K}^{\leftarrow u}_{\mathbf{w}} = \frac{\sum_{v \in \text{in}(u)} \mathcal{K}^v_{\mathbf{w}}}{|\text{in}(u)|}, \quad \mathcal{K}^{\leftarrow u} = \frac{\sum_{v \in \text{in}(u)} \mathcal{K}^v}{|\text{in}(u)|}.$$

For a matrix $\mathcal{K} \in \mathcal{M}_+$ and a function $f : \mathcal{M}_+ \to \mathbb{R}$, we denote

$$f^p(\mathcal{K}) = \begin{pmatrix} f\begin{pmatrix} \mathcal{K}_{11} & \mathcal{K}_{11} \\ \mathcal{K}_{11} & \mathcal{K}_{11} \end{pmatrix} & f(\mathcal{K}) \\ f(\mathcal{K}) & f\begin{pmatrix} \mathcal{K}_{22} & \mathcal{K}_{22} \\ \mathcal{K}_{22} & \mathcal{K}_{22} \end{pmatrix} \end{pmatrix}$$

Note that $\mathcal{K}^u = \bar{\sigma}_u^p(\mathcal{K}^{\leftarrow u})$. We say that a node $u \in \mathcal{S}$, is *well-initialized* if

$$\|\mathcal{K}_{\mathbf{w}}^u - \mathcal{K}^u\|_\infty \le \epsilon \frac{B_{\mathrm{depth}(u)}}{B_{\mathrm{depth}(\mathcal{S})}} . \tag{11}$$

Here, we use the convention that $B_0 = 0$. It is enough to show that with probability of at least $\ge 1 - \delta$ all nodes are well-initialized. We first note that input nodes are well-initialized by construction since $\mathcal{K}_{\mathbf{w}}^u = \mathcal{K}^u$. Next, we show that given that all incoming nodes for a certain node are well-initialized, then w.h.p. the node is well-initialized as well.

**Claim 3.** *Assume that all the nodes in* $\mathrm{in}(u)$ *are well-initialized. Then, the node $u$ is well-initialized with probability of at least* $1 - \frac{\delta}{|\mathcal{S}|}$.

*Proof.* It is easy to verify that $\mathcal{K}_{\mathbf{w}}^u$ is the empirical covariance matrix of $r$ independent variables distributed according to $(\sigma(X), \sigma(Y))$ where $(X, Y) \sim \mathrm{N}(0, \mathcal{K}_{\mathbf{w}}^{\leftarrow u})$. Given the assumption that all nodes incoming to $u$ are well-initialized, we have,

$$
\begin{aligned}
\|\mathcal{K}_{\mathbf{w}}^{\leftarrow u} - \mathcal{K}^{\leftarrow u}\|_\infty &= \left\| \frac{\sum_{v \in \mathrm{in}(v)} \mathcal{K}_{\mathbf{w}}^v}{|\mathrm{in}(v)|} - \frac{\sum_{v \in \mathrm{in}(v)} \mathcal{K}^v}{|\mathrm{in}(v)|} \right\|_\infty \\
&\le \frac{1}{|\mathrm{in}(v)|} \sum_{v \in \mathrm{in}(v)} \|\mathcal{K}_{\mathbf{w}}^v - \mathcal{K}^v\|_\infty \\
&\le \epsilon \frac{B_{\mathrm{depth}(u)-1}}{B_{\mathrm{depth}(\mathcal{S})}} .
\end{aligned}
\tag{12}
$$

Further, since $\epsilon \le \gamma$ then $\mathcal{K}_{\mathbf{w}}^{\leftarrow u} \in \mathcal{M}_+^\gamma$. Using the fact that $\sigma_u$ is $(\alpha, \beta, \gamma)$-decent and that $r \ge \frac{2\alpha^2 B_{\mathrm{depth}(\mathcal{S})}^2 \log\left(\frac{8|\mathcal{S}|}{\delta}\right)}{\epsilon^2}$, we get that w.p. of at least $1 - \frac{\delta}{|\mathcal{S}|}$,

$$\|\mathcal{K}_{\mathbf{w}}^u - \bar{\sigma}_u^p(\mathcal{K}_{\mathbf{w}}^{\leftarrow u})\|_\infty \le \frac{\epsilon}{B_{\mathrm{depth}(\mathcal{S})}} . \tag{13}$$

Finally, using (12) and (13) along with the fact that $\bar{\sigma}$ is $\beta$-Lipschitz, we have

$$
\begin{aligned}
\|\mathcal{K}_{\mathbf{w}}^u - \mathcal{K}^u\|_\infty &= \|\mathcal{K}_{\mathbf{w}}^u - \bar{\sigma}_u^p(\mathcal{K}^{\leftarrow u})\|_\infty \\
&\le \|\mathcal{K}_{\mathbf{w}}^u - \bar{\sigma}_u^p(\mathcal{K}_{\mathbf{w}}^{\leftarrow u})\|_\infty + \|\bar{\sigma}_u^p(\mathcal{K}_{\mathbf{w}}^{\leftarrow u}) - \bar{\sigma}_u^p(\mathcal{K}^{\leftarrow u})\|_\infty \\
&\le \frac{\epsilon}{B_{\mathrm{depth}(\mathcal{S})}} + \beta \|\mathcal{K}_{\mathbf{w}}^{\leftarrow u} - \mathcal{K}^{\leftarrow u}\|_\infty \\
&\le \frac{\epsilon}{B_{\mathrm{depth}(\mathcal{S})}} + \beta\epsilon \frac{B_{\mathrm{depth}(u)-1}}{B_{\mathrm{depth}(\mathcal{S})}} = \epsilon \frac{B_{\mathrm{depth}(u)}}{B_{\mathrm{depth}(\mathcal{S})}} . \quad \square
\end{aligned}
$$

We are now ready to conclude the proof. Let $u_1, \ldots, u_{|\mathcal{S}|}$ be an ordered list of the nodes in $\mathcal{S}$ in accordance to their depth, starting with the shallowest nodes, and ending with the output node. Denote by $A_q$ the event that $u_1, \ldots, u_q$ are well-initialized. We need to show that $\Pr(A_{|\mathcal{S}|}) \ge 1 - \delta$. We do so using an induction on $q$ for the inequality $\Pr(A_q) \ge 1 - \frac{q\delta}{|\mathcal{S}|}$. Indeed, for $q = 1, \ldots, n$, $u_q$ is an input node and $\Pr(A_q) = 1$. Thus, the base of the induction hypothesis holds. Assume that $q > n$. By Claim (3) we have that $\Pr(A_q | A_{q-1}) \ge 1 - \frac{\delta}{|\mathcal{S}|}$. Finally, from the induction hypothesis we have,

$$\Pr(A_q) \ge \Pr(A_q | A_{q-1}) \Pr(A_{q-1}) \ge \left(1 - \frac{\delta}{|\mathcal{S}|}\right)\left(1 - \frac{(q-1)\delta}{|\mathcal{S}|}\right) \ge 1 - \frac{q\delta}{|\mathcal{S}|} . \quad \square$$

## D.3 Proofs of Thms. 5 and 6

Theorems 5 and 6 follow from using the following lemma combined with Theorems 3 and 4. When we apply the lemma, we always focus on the special case where one of the kernels is constant w.p. 1.

**Lemma 18.** *Let $\mathcal{D}$ be a distribution on $\mathcal{X} \times \mathcal{Y}$, $\ell : \mathbb{R} \times \mathcal{Y} \to \mathbb{R}$ be an $L$-Lipschitz loss, $\delta > 0$, and $\kappa_1, \kappa_2 : \mathcal{X} \times \mathcal{X} \to \mathbb{R}$ be two independent random kernels sample from arbitrary distributions. Assume that the following properties hold.*

- *For some $C > 0$, $\forall \mathbf{x} \in \mathcal{X}$, $\kappa_1(\mathbf{x}, \mathbf{x}), \kappa_2(\mathbf{x}, \mathbf{x}) \leq C$.*

- *$\forall \mathbf{x}, \mathbf{y} \in \mathcal{X}$, $\Pr_{\kappa_1, \kappa_2} \left( |\kappa_1(\mathbf{x}, \mathbf{y}) - \kappa_2(\mathbf{x}, \mathbf{y})| \geq \epsilon \right) \leq \tilde{\delta}$ for $\tilde{\delta} < c_2 \frac{\epsilon^2 \delta}{C^2 \log^2\left(\frac{1}{\delta}\right)}$ where $c_2 > 0$ is a universal constant.*

*Then, w.p. $\geq 1 - \delta$ over the choices of $\kappa_1, \kappa_2$, for every $f_1 \in \mathcal{H}_{\kappa_1}^M$ there is $f_2 \in \mathcal{H}_{\kappa_2}^{\sqrt{2}M}$ such that $\mathcal{L}_{\mathcal{D}}(f_2) \leq \mathcal{L}_{\mathcal{D}}(f_1) + \sqrt{\epsilon} 4LM$.*

To prove the above lemma, we state another lemma below followed by a basic measure concentration result.

**Lemma 19.** *Let $\mathbf{x}_1, \ldots, \mathbf{x}_m \in \mathbb{R}^d$, $\mathbf{w}^* \in \mathbb{R}^d$ and $\epsilon > 0$. There are weights $\alpha_1, \ldots, \alpha_m$ such that for $\mathbf{w} := \sum_{i=1}^{m} \alpha_i \mathbf{x}_i$ we have,*

- *$\mathcal{L}(\mathbf{w}) := \frac{1}{m} \sum_{i=1}^{m} |\langle \mathbf{w}, \mathbf{x}_i \rangle - \langle \mathbf{w}^*, \mathbf{x}_i \rangle| \leq \epsilon$*

- *$\sum_i |\alpha_i| \leq \frac{\|\mathbf{w}^*\|^2}{\epsilon}$*

- *$\|\mathbf{w}\| \leq \|\mathbf{w}^*\|$*

*Proof.* Denote $M = \|\mathbf{w}^*\|$, $C = \max_i \|\mathbf{x}_i\|$, and $y_i = \langle \mathbf{w}^*, \mathbf{x}_i \rangle$. Suppose that we run stochastic gradient decent on the sample $\{(\mathbf{x}_1, y_1), \ldots, (\mathbf{x}_m, y_m)\}$ w.r.t. the loss $\mathcal{L}(\mathbf{w})$, with learning rate $\eta = \frac{\epsilon}{C^2}$, and with projections onto the ball of radius $M$. Namely, we start with $\mathbf{w}_0 = 0$ and at each iteration $t \geq 1$, we choose at random $i_t \in [m]$ and perform the update,

$$\tilde{\mathbf{w}}_t = \begin{cases} \mathbf{w}_{t-1} - \eta \mathbf{x}_{i_t} & \langle \mathbf{w}_{t-1}, \mathbf{x}_{i_t} \rangle \geq y_{i_t} \\ \mathbf{w}_{t-1} + \eta \mathbf{x}_{i_t} & \langle \mathbf{w}_{t-1}, \mathbf{x}_{i_t} \rangle < y_{i_t} \end{cases}$$

$$\mathbf{w}_t = \begin{cases} \tilde{\mathbf{w}}_t & \|\tilde{\mathbf{w}}_t\| \leq M \\ \frac{M \tilde{\mathbf{w}}_t}{\|\tilde{\mathbf{w}}_t\|} & \|\tilde{\mathbf{w}}_t\| > M \end{cases}$$

After $T = \frac{M^2 C^2}{\epsilon^2}$ iterations the loss in expectation would be at most $\epsilon$ (see for instance Chapter 14 in [43]). In particular, there exists a sequence of at most $\frac{M^2 C^2}{\epsilon^2}$ gradient steps that attains a solution $\mathbf{w}$ with $\mathcal{L}(\mathbf{w}) \leq \epsilon$. Each update adds or subtracts $\frac{\epsilon}{C^2} \mathbf{x}_i$ from the current solution. Hence $\mathbf{w}$ can be written as a weighted sum of $\mathbf{x}_i$'s where the sum of each coefficient is at most $T \frac{\epsilon}{C^2} = \frac{M^2}{\epsilon}$. $\square$

**Theorem 20** (Bartlett and Mendelson [8]). *Let $\mathcal{D}$ be a distribution over $\mathcal{X} \times \mathcal{Y}$, $\ell : \mathbb{R} \times \mathcal{Y} \to \mathbb{R}$ a 1-Lipschitz loss, $\kappa : \mathcal{X} \times \mathcal{X} \to \mathbb{R}$ a kernel, and $\epsilon, \delta > 0$. Let $S = \{(\mathbf{x}_1, y_1), \ldots, (\mathbf{x}_m, y_m)\}$ be i.i.d. samples from $\mathcal{D}$ such that $m \geq c \frac{M^2 \max_{\mathbf{x} \in \mathcal{X}} \kappa(\mathbf{x}, \mathbf{x}) + \log\left(\frac{1}{\delta}\right)}{\epsilon^2}$ where $c$ is a constant. Then, with probability of at least $1 - \delta$ we have,*

$$\forall f \in \mathcal{H}_{\kappa}^M, \ |\mathcal{L}_{\mathcal{D}}(f) - \mathcal{L}_S(f)| \leq \epsilon.$$

*Proof.* (of Lemma 18) By rescaling $\ell$, we can assume w.l.o.g that $L = 1$. Let $\epsilon_1 = \sqrt{\epsilon} M$ and $S = \{(\mathbf{x}_1, y_1), \ldots, (\mathbf{x}_m, y_m)\} \sim \mathcal{D}$ be i.i.d. samples which are independent of the choice of $\kappa_1, \kappa_2$. By Theorem 20, for a large enough constant $c$, if $m = c \frac{CM^2 \log\left(\frac{1}{\delta}\right)}{\epsilon_1^2} = c \frac{C \log\left(\frac{1}{\delta}\right)}{\epsilon}$, then w.p. $\geq 1 - \frac{\delta}{2}$ over the choice of the samples we have,

$$\forall f \in \mathcal{H}_{\kappa_1}^M \cup \mathcal{H}_{\kappa_2}^{\sqrt{2}M}, \ |\mathcal{L}_{\mathcal{D}}(f) - \mathcal{L}_S(f)| \leq \epsilon_1 \tag{14}$$

Now, if we choose $c_2 = \frac{1}{2c^2}$ then w.p. $\geq 1 - m^2 \tilde{\delta} \geq 1 - \frac{\delta}{2}$ (over the choice of the examples and the kernel), we have that

$$\forall i, j \in [m], |\kappa_1(\mathbf{x}_i, \mathbf{x}_j) - \kappa_2(\mathbf{x}_i, \mathbf{x}_j)| < \epsilon. \tag{15}$$

In particular, w.p. $\geq 1 - \delta$ (14) and (15) hold and therefore it suffices to prove the conclusion of the theorem under these conditions. Indeed, let $\Psi_1, \Psi_2 : \mathcal{X} \to \mathcal{H}$ be two mapping from $\mathcal{X}$ to a Hilbert space $\mathcal{H}$ so that $\kappa_i(\mathbf{x}, \mathbf{y}) = \langle \Psi_i(\mathbf{x}), \Psi_i(\mathbf{y}) \rangle$. Let $f_1 \in \mathcal{H}_{\kappa_1}^M$. By lemma 19 there are $\alpha_1, \ldots, \alpha_m$ so that for the vector $\mathbf{w} = \sum_{i=1}^m \alpha_1 \Psi_1(\mathbf{x}_i)$ we have

$$\frac{1}{m} \sum_{i=1}^m |\langle \mathbf{w}, \Psi_1(\mathbf{x}_i) \rangle - f_1(\mathbf{x}_i)| \leq \epsilon_1, \quad \|\mathbf{w}\| \leq M, \tag{16}$$

and

$$\sum_{i=1}^m |\alpha_i| \leq \frac{M^2}{\epsilon_1}. \tag{17}$$

Consider the function $f_2 \in \mathcal{H}_2$ defined by $f_2(\mathbf{x}) = \sum_{i=1}^m \alpha_1 \langle \Psi_2(\mathbf{x}_i), \Psi_2(\mathbf{x}) \rangle$. We note that

$$
\begin{aligned}
\|f_2\|_{\mathcal{H}_{k_2}}^2 &\leq \left\| \sum_{i=1}^m \alpha_i \Psi_2(\mathbf{x}_i) \right\|^2 \\
&= \sum_{i,j=1}^m \alpha_i \alpha_j \kappa_2(\mathbf{x}_i, \mathbf{x}_j) \\
&\leq \sum_{i,j=1}^m \alpha_i \alpha_j \kappa_1(\mathbf{x}_i, \mathbf{x}_j) + \epsilon \sum_{i,j=1}^m |\alpha_i \alpha_j| \\
&= \|\mathbf{w}\|^2 + \epsilon \left( \sum_{i=1}^m |\alpha_i| \right)^2 \\
&\leq M^2 + \epsilon \frac{M^4}{\epsilon_1^2} = 2M^2.
\end{aligned}
$$

Denote by $\tilde{f}_1(\mathbf{x}) = \langle \mathbf{w}, \Psi_1(\mathbf{x}) \rangle$ and note that for every $i \in [m]$ we have,

$$
\begin{aligned}
|\tilde{f}_1(\mathbf{x}_i) - f_2(\mathbf{x}_i)| &= \left| \sum_{j=1}^m \alpha_j \left( \kappa_1(\mathbf{x}_i, \mathbf{x}_j) - \kappa_2(\mathbf{x}_i, \mathbf{x}_j) \right) \right| \\
&\leq \epsilon \sum_{i=1}^m |\alpha_i| \leq \epsilon \frac{M^2}{\epsilon_1} = \epsilon_1.
\end{aligned}
$$

Finally, we get that,

$$
\begin{aligned}
\mathcal{L}_{\mathcal{D}}(f_2) &\leq \mathcal{L}_S(f_2) + \epsilon_1 \\
&= \frac{1}{m} \sum_{i=1}^m \ell(f_2(\mathbf{x}_i), y_i) + \epsilon_1 \\
&\leq \frac{1}{m} \sum_{i=1}^m \ell\left(\tilde{f}_1(\mathbf{x}_i), y_i\right) + \epsilon_1 + \epsilon_1 \\
&\leq \frac{1}{m} \sum_{i=1}^m \ell(f_1(\mathbf{x}_i), y_i) + |\tilde{f}_1(\mathbf{x}_i) - f_1(\mathbf{x}_i)| + 2\epsilon_1 \\
&\leq \frac{1}{m} \sum_{i=1}^m \ell(f_1(\mathbf{x}_i), y_i) + 3\epsilon_1 \\
&\leq \mathcal{L}_S(f_1) + 3\epsilon_1 \leq \mathcal{L}_{\mathcal{D}}(f_1) + 4\epsilon_1,
\end{aligned}
$$

which concludes the proof. $\qquad \square$