[Reviews · NeurIPS 2016]

Reviewer 1

Summary

The article introduces a new computational model called computation skeleton. It makes it possible to develop a duality between feed-forward neural networks and compositional kernel Hilbert spaces. The main results established are lower bounds on the replication parameter r.

Qualitative Assessment

This contribution to the theory of feed-forward neural networks, through the introduction of a new computational model, the computation skeleton, seems both original and promising. The reader who is not very familiar with deep neural networks (this is my case) will find it harder to understand the paper. The main weakness of the presentation rests in the fact that the motivations for some of the major hypotheses made (see Section 2) are to be found in the supplemnetary material. To my mind, the 9-page paper should be more self-contained. Furthermore, there are quite a few typos.

Confidence in this Review

2-Confident (read it all; understood it all reasonably well)


Reviewer 2

Summary

The authors give a dual relationship between neural networks and compositional reproducing kernel Hilbert spaces. This is based on an introduced notion of computation skeleton. The corresponding optimization problems are also considered.

Qualitative Assessment

The duality is well described and the notion of computation skeleton is well presented. The theoretical bounds involving the optimization problems seem reasonable, but I did not check the proofs. However, no numerical experiments are provided. It would be good to demonstrate a new concept by some experiments.

Confidence in this Review

2-Confident (read it all; understood it all reasonably well)


Reviewer 3

Summary

The paper introduced a concept of computation skeleton associated to neural networks and defined a composition kernel. This kernel can be approximated by the empirical kernel induced by an random initialization of the neural network.

Qualitative Assessment

This paper discussed the relationship between several concepts associated to neural networks, especially the relation of composite kernel and empirical kernel. Although the results looks somewhat novel and interesting, the main problem of the paper is lack of discussion on why these results are important to understand the neural network learning either theoretically or empirically.

Confidence in this Review

1-Less confident (might not have understood significant parts)


Reviewer 4

Summary

The authors introduce the notion of computational skeleton and show how a skeleton can be related to a RKHS (more precisely, to a compositional kernel) through the concept of "dual activation". They also introduce a class of neural networks, called "realisation of a skeleton", defined through a given number of replications of a computational skeleton and a given number a of outputs (r and k, respectively). The main results of the paper consist in analysing in what sense the representation generated by a random initialisation of such a neural network (i.e., a skeleton realisation) approximates the compositional kernel related to the underlying skeleton.

Qualitative Assessment

# Some typos/remarks: -At line 154, do the authors mean "any continuous PSD function"? -Two typos at lines 171 (y instead of x') and 177 ("of in the induced space"). -I am not sure about the meaning of the "2-norm" mentioned at lines 173 and 184, maybe the author should specify it. -At line 190, S is a computational skeleton, and k_{S} is a compositional kernel. -In Lemma 11 (appendix), I do not really understand what does b_{|A|} mean. #Questions (maybe not founded...): -Is it possible to interpret the fact that "for a random activation, the empirical kernel k_w approximates the kernel k_S", as a kind of "law of large numbers", in the sense: "the replication of the computation skeleton counterbalances the fact that the weight of the network are random"? -What kind of impact the operation of "learning the weights of the network" could have, for instance, on the relation between k_w and k_S (and on the related Hilbert spaces)? -Is it possible, in view of Thm. 5 and 6, to draw a parallel between "learning with a skeleton realisation" and "learning with a compositional kernel"? ########################################### ### After rebuttal My major concern about the paper is the lack of discussion on "why these results may be important or interesting". In fact, there is almost no discussion at all about the implication of the results of the paper. The notions of "computational skeleton" and "realisation of a skeleton" seem relatively interesting and aim at generalising the kernel construction proposed in [13] and [29]. This "construction" is in my view the main contribution of the paper. On a theoretical point of view and from my understanding, the main results of the paper are Thm 3. and 4. However, these results are not "so surprising", since, as confirmed by the authors, they can be interpreted as a kind of "law of large number": the random activation of the network is "counterbalanced" by the replication of the skeleton. Note that there is a big typo in eq. (12) of the supplementary material. As indicated by Table 1, some results of this type are already known. From my understanding, Thm 5 and 6 are in fact just "advanced corollaries" of Thm 3. and 4 (as indicated at line 586 of the appendix, with a typo). In their feedback, the authors indicate (about Thm 5 and 6) that: "this shows that training the final layer is essentially equivalent to learning with the corresponding kernel". However, for "kernel-based learning" (line 82), the authors deal with the classical "loss + regularisation" formulation, while classically, "only loss" is considered for "Neural Network-based learning" (line 74); so that I do not really understand how "training the final layer" could be sensitive to "the size of the ball in the RKHS". In my view and from my understanding, these results are about "the difference between learning with K_S or K_w in the classical framework of kernel-based learning", so that I do not really agree with the answer of the authors. Another "drawback" of the approach is, in my opinion, that the input space is restricted to be of "hypersphere-type". Finally, the paper is quite difficult to read and its construction may, in my opinion, be improved in order to try to clarify and simplify the paper.

Confidence in this Review

2-Confident (read it all; understood it all reasonably well)


Reviewer 5

Summary

The paper introduces a new connection between neural networks and kernel methods. The paper introduces the concept of the computational skeleton representing a neural network, then defines a kernel function from this which may be viewed as a mean kernel when weights are drawn from a normal distribution. The paper investigates how well this kernel approximates the kernel induced by a skeleton with randomly selected weights, and uses this for a basis for discussion of how deep networks are able to perform so well when much theoretical work often predicts that they will barely work better than random.

Qualitative Assessment

(caveat: while I have a reasonably strong background in kernel methods my understanding of (deep) neural networks is less well developed). The paper presents an interesting and (to the best of my knowledge) new approach to investigating the performance of neural networks in terms of kernel methods. The results appear sound and have interesting implications for both fields. Minor points: - I'm not sure that citing wikipedia is advisable (line 341). - Lemma 11: it is unclear what b is in this lemma. Does it refers to the Taylor expansion of mu? I assume that |A| refers to the cardinality of A?

Confidence in this Review

2-Confident (read it all; understood it all reasonably well)


Reviewer 6

Summary

The paper expands previous interpretations of deep learning as a kernel machine which were limited to two-layer fully connected networks to any kind or architecture. In order to do so, the concept of computation skeleton is introduced. Using those skeletons, it proves that if the network fulfils a set of conditions on the number of hidden neurons and its depth and on the type of activation function, the resulting network approximates the kernel generated by the skeleton of that network.

Qualitative Assessment

First of all, I must admit that I probably lack the expertise to fully grasp the content of this paper, yet I will try to contribute with my point of view as kernel svm and deep learning newbie researcher. Although I honestly failed to understand the main results, I found very exciting how the authors model neural networks as computation skeletons and then interpret them as kernels. My personal opinion is that one of the main drawbacks of deep learning is the lack of mathematical foundation for the architectures, which makes finding the optimal topology to be a costly and poor trial and error process. It seems to me that most of the research to solve this problem comes the probabilistic approach, so a work which uses the kernel framework is very interesting to me. Conclusively, the modelization of neural networks as computation skeletons and kernels proposed by the authors alone justify the high evaluation I give, to my humble opinion. However, further review of the main results section is required. Please accept my apologies for not being of more help.

Confidence in this Review

1-Less confident (might not have understood significant parts)